# HIV-1 Vpr orchestrates ciTRAN upregulation through TGF-β induction

Aditi Choudhary, Katyayani Mallick, Rishikesh Dalavi, Ajit Chande ⓘ *

Molecular Virology Laboratory, Department of Biological Sciences, Indian Institute of Science Education and Research (IISER) Bhopal, Bhopal, India

* ajitg@iiserb.ac.in

## Abstract

Circular RNA (circRNA) expression is widespread in immune cells infected by HIV-1, but the crosstalk between circRNA expression and various cellular signaling pathways remains elusive. Here, we report that HIV-1 Vpr can induce the production of TGF-β during infection, which we linked to the upregulation of ciTRAN, a proviral circRNA encoded by *SMARCA5*. Consistent with this finding, we observed that the essential intracellular TGF-β receptor signaling component SMAD2/3 was recruited to the *SMARCA5* promoter in a Vpr-dependent manner. *SMARCA5* promoter analysis and functional assays further revealed that SAMD2/3 binding motif in the *SMARCA5* promoter is crucial for ciTRAN upregulation. Notably, in response to treatment with DNA-damaging agents or the exogenous addition of recombinant TGF-β, the TGF-SMAD axis upregulated the expression of ciTRAN as well as the parental *SMARCA5* mRNA. Regardless, the QKI protein was necessary for ciTRAN biogenesis. Finally, pharmacological targeting or genetic ablation of *TGFBR1* can reduce the ability of Vpr to promote the expression of ciTRAN and viral genes. These results highlight the TGF-β-mediated regulation of ciTRAN expression which may play a role in modulating HIV-1 replication.

## Author summary

HIV-1 remains a global challenge, driving the exploration of novel strategies to combat the virus. Emerging evidence suggests that HIV-1 infection is also associated with changes in circular RNA profiles in CD4+ T cells. By an unknown mechanism, the accessory protein Vpr encoded by the primate lentivirus was shown to induce the expression of a *SMARCA5*-encoded proviral circular RNA ciTRAN (circSMARCA5). Here, we show that HIV-1 Vpr stimulates TGF-β production, orchestrating the upregulation of ciTRAN. Accordingly, the key signaling proteins (SMAD2/3) downstream of TGF-β receptors bind to the *SMARCA5* promoter to boost ciTRAN production in the presence of Vpr. The experiments also

**Data availability statement:** All data associated with this manuscript are included in the figures and Supporting information.

**Funding:** This work was supported by the DBT/Wellcome Trust India Alliance Fellowship (IA/I/18/2/504006), and by European Molecular Biology Organization (#5753), Lady Tata Memorial Trust Young Scientist Award, Indian Council of Medical Research (EM/dev//CAR/-2024-01-0113/F2/2024) and Anusandhan National Research Foundation (ANRF) grant # CRG/2023/000720 (to A.Cha). The funders had no role in study design, data collection and analysis, decision to publish, or preparation of the manuscript.

**Competing interests:** The authors have declared that no competing interests exist.

indicate that treatment with chemotherapeutic DNA damaging agents or exogenous addition of recombinant TGF-β can upregulate ciTRAN. However, blocking the TGF-β signaling pathway by small molecules or depleting its components by genetic means diminished the ability of Vpr to stimulate ciTRAN and the viral gene expression. These findings highlight the TGF-β control of ciTRAN expression and suggest potential ways to block circular RNA expression during HIV-1 infection using small molecules.

## Introduction

Circular RNAs (circRNAs) are an important subclass of RNAs that are widespread in metazoans [1–3]. In contrast to linear RNAs, which are produced by canonical splicing, circRNAs are generated by backsplicing, wherein a 5' splice donor links to a 3' splice site upstream, generating a circular species of RNA resistant to exonuclease-mediated decay [4–6]. CircRNAs are evolutionarily conserved and developmentally regulated. Moreover, circRNAs display cell-specific expression patterns, are intrinsically stable, and can regulate various cellular processes [7,8]. In addition to the early studies revealing microRNA sponging as a primary function of circRNAs [9,10], recent studies reveal many other roles, such as regulators of gene expression, serving as templates for protein synthesis, and modulation of viral infections [11].

The pro- and anti-viral roles of circRNAs derived from hosts and different viruses are becoming evident [12–16]. Retroviruses are no exception. A host-encoded circRNA co-opted by HIV-1 to promote its transcription was recently reported highlighting the importance of circRNA in retroviral pathogenesis [17]. Complex retroviruses such as HIV-1 are important global public health concerns, and the addition of circRNA as a new player in the host-virus arms race further demonstrates the complexity of host-virus interactions [11]. In particular, the accessory proteins encoded by HIV-1 play crucial roles in natural infection [18], and how accessory proteins can also function by hijacking the host circRNA network is crucial for obtaining mechanistic insights into viral pathogenesis. The viral accessory protein Vpr was shown to induce the expression of the *SMARCA5*-encoded circRNA ciTRAN in the previous study [17]. In the infected cells, this primate lentivirus protein shows discrete activities: it can modulate DNA damage, replication stalling, homologous recombination (HR) repair, and cell cycle arrest [19–22]. The DNA damage response (DDR) pathway is modulated by the accessory protein by promoting DNA damage and regulating the functions of proteins involved in DDR signaling [23,24]. It has been canonically shown to interact with DDB-CUL4A-associated factor 1 (DCAF1), regulating functions like cell cycle arrest and proteasome-dependent target protein degradation, which further assists HIV-1 replication [25–27]. Classified as an immediate-early protein, Vpr gets incorporated within the particles [28–30], and can also regulate various cell signaling pathways by modulating viral and host gene expression [31–36]. However, the crosstalk between various activities of Vpr, its impact on cellular pathways, and circular RNA expression is incompletely understood.

PLOS Pathogens

Among various cellular pathways, the transforming growth factor (TGF-β) pathway is essential for regulating key cellular processes, including proliferation, differentiation, and apoptosis [37]. This pathway is activated when TGF-β ligands bind to type I and type II serine/threonine kinase receptors, causing the phosphorylation of SMAD proteins [38]. These phosphorylated SMAD proteins then migrate to the nucleus to influence gene expression. TGF-β signaling is also closely linked with DDR pathways, crucial for maintaining genomic stability. Upon DNA damage, TGF-β can modulate the DDR by affecting the expression of genes involved in cell cycle arrest, DNA repair, and apoptosis [39]. For example, TGF-β promotes the repair of DNA double-strand breaks via the nonhomologous end joining (NHEJ) and homologous recombination (HR) pathways [40]. Additionally, TGF-β signaling can induce cell cycle arrest, providing cells with more time to repair DNA damage before division. Dysregulation of TGF-β pathway is often associated with cancer, as it can influence DNA repair mechanisms and increase genomic instability, contributing to tumor progression and chemotherapy resistance [41,42]. TGF-β signaling is also shown to promote CCR5- tropic HIV-1 infection in resting and activated memory CD4[+] T cells in SMAD3- dependent manner and blockade of TGF-β was shown to drive transitional effector phenotype in T cells [43,44]. Higher plasma levels of TGF-β are implicated in disease progression, impaired immune functions, increased viral replication, and CD4[+] T-cell depletion, suggesting TGF-β as a crucial pathogenic mediator [45–50]. However, the viral mediators of increased TGF-β in cells infected by HIV-1 remain elusive.

Given the parallel effects of Vpr and TGF-β on HIV replication, we hypothesized that ciTRAN induction by HIV-1 Vpr is intricately linked to TGF-β.

## Results

### ciTRAN upregulation is associated with DNA damage

To obtain detailed insights into how the expression of ciTRAN is induced by HIV-1 Vpr, we employed a panel of mutants that genetically separated the activities of the accessory protein (Figs 1A, S1A). JTAg T cells challenged 30h before with lentiviral vector (LV) particles encapsidating Vpr protein (S1B Fig) were harvested, and the ability of Vpr to induce RNaseR-resistant ciTRAN expression was assessed from the total RNA using a set of divergent primers as reported earlier [17] (S1C Fig). In conditions where JTAg cells received indicated Vprs packaged into virions (S1B Fig), the cells receiving Vpr Q65R mutant showed comparable ciTRAN expression to that of cells without Vpr, whereas the other mutants increased the level of ciTRAN to varying extents (4-to-6-fold) (Fig 1B). The hemagglutinin (HA) tagged Vprs were also detectable from the lysates of JTAg cells by western blotting (Fig 1B: bottom panel). The Q65R mutant was reported to be defective in inducing the DNA damage response, in addition to other activities like arresting the cell cycle at the G2 phase, stalling the replication fork, repressing HR and binding to DCAF1 [22,23,51,52]. Whereas the W54R mutant was shown defective in binding to UNG2 but retained all the above functions, including cell cycle arrest [53]. The S79A mutant, proficient in inducing the DDR but failed to arrest the cell cycle or stall the replication fork and HR repression [21,26,54], also increased ciTRAN levels. The analysis of DNA damage by tail profile via an alkaline comet assay further revealed that WT Vpr, W54R, S79A, and R80A were inducing DNA damage; however, the Q65R mutant consistently failed to induce DNA damage under these experimental conditions (Fig 1C). We also confirmed the DNA damage activation marker γH2AX by immunofluorescence assay (S1D Fig) as well as its phosphorylation status by western blotting (S1E Fig) for WT and mutant Vprs indicating the activation of DDR pathway. Subsequent analysis of endpoints from these two orthogonal assays revealed that the extent of DNA damage by Vpr (expressed as % tail length) correlates with ciTRAN upregulation (Fig 1D). Thus, we next examined whether DNA damaging agents alone can upregulate ciTRAN in the absence of infection. Interestingly, ciTRAN expression induction by DNA-damaging agents (Etoposide (ETO) and Doxorubicin (Dox)), which cleave DNA randomly, was reminiscent of virion-incorporated Vpr action (Fig 1E). These experiments suggested that DNA damage is associated with ciTRAN upregulation.

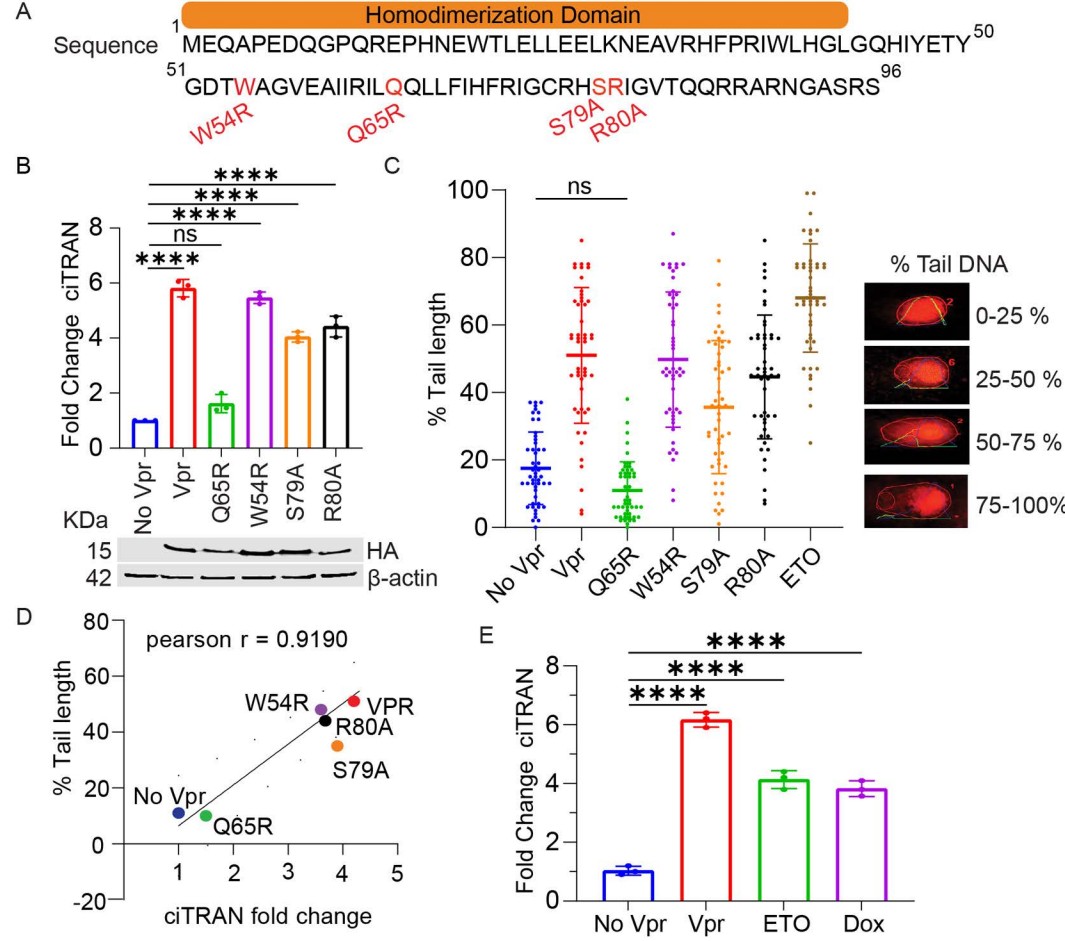

**Fig 1. ciTRAN upregulation upon DNA damage.** (A) Schematic representation of HIV-1 Vpr protein (uniport ID P05928) and highlighted amino acids that were mutated. (B) ciTRAN levels from JTAg cells, transduced with lentiviral vectors (LV) encapsidating the indicated Vpr proteins, were quantified by qRT-PCR after 30 h. Lower panel: Immunoblotting from the lysates of LV producer cells showing corresponding Hemagglutinin (HA)-tagged Vpr proteins. (C) Alkaline comet assay indicating the extent of DNA damage in JTAg cells challenged with LV-packaged WT or mutant Vpr proteins, n =50±SD. The bars represent the median with the interquartile range. Right panel, visual representation of the four degrees of damage measured by percent tail DNA. Intensity profiles, lines, and numbers on the images were automatically generated by the *OpenComet* plug-in for ImageJ. (D) Correlation of percent tail-length and fold change in ciTRAN induction has been shown by plotting the mean values for each condition. (E) ciTRAN levels after Vpr (+/-) LV transduction of JTAg cells as in (B) or after 6hr treatment with 10µM Etoposide (ETO) or Doxorubicin (Dox). The two-tailed Student's t-test (unpaired) or one-way ANOVA with Dunnett's Multiple comparison test was used to assess the significance between two or more groups, ns= non-significant, *p < 0.05, **p < 0.01, ***p < 0.001 and ****p < 0.0001.

### ciTRAN upregulation is not mediated by DCAF1 or the canonical DDR pathway proteins and is independent of cell cycle arrest

DCAF1 functions as a substrate recognition adaptor for CRL4 protein complexes, mediating protein ubiquitination and regulating cell-cycle progression. We further investigated whether Vpr exploits DCAF1 to induce ciTRAN during HIV-1 infection. Since Vpr Q65R/A fails to recruit DCAF1 and cannot mediate target protein degradation (S1A Fig) [55,56], it remained a possibility. To test this, *DCAF1* knockdown was performed using shRNA (S2A Fig) to evaluate its effect on ciTRAN induction (Fig 2A). Additionally, a mutant selectively deficient in DCAF1 binding Vpr H71R was generated that retained other functions inducing DNA damage, causing cell cycle arrest, and homologous recombination suppression

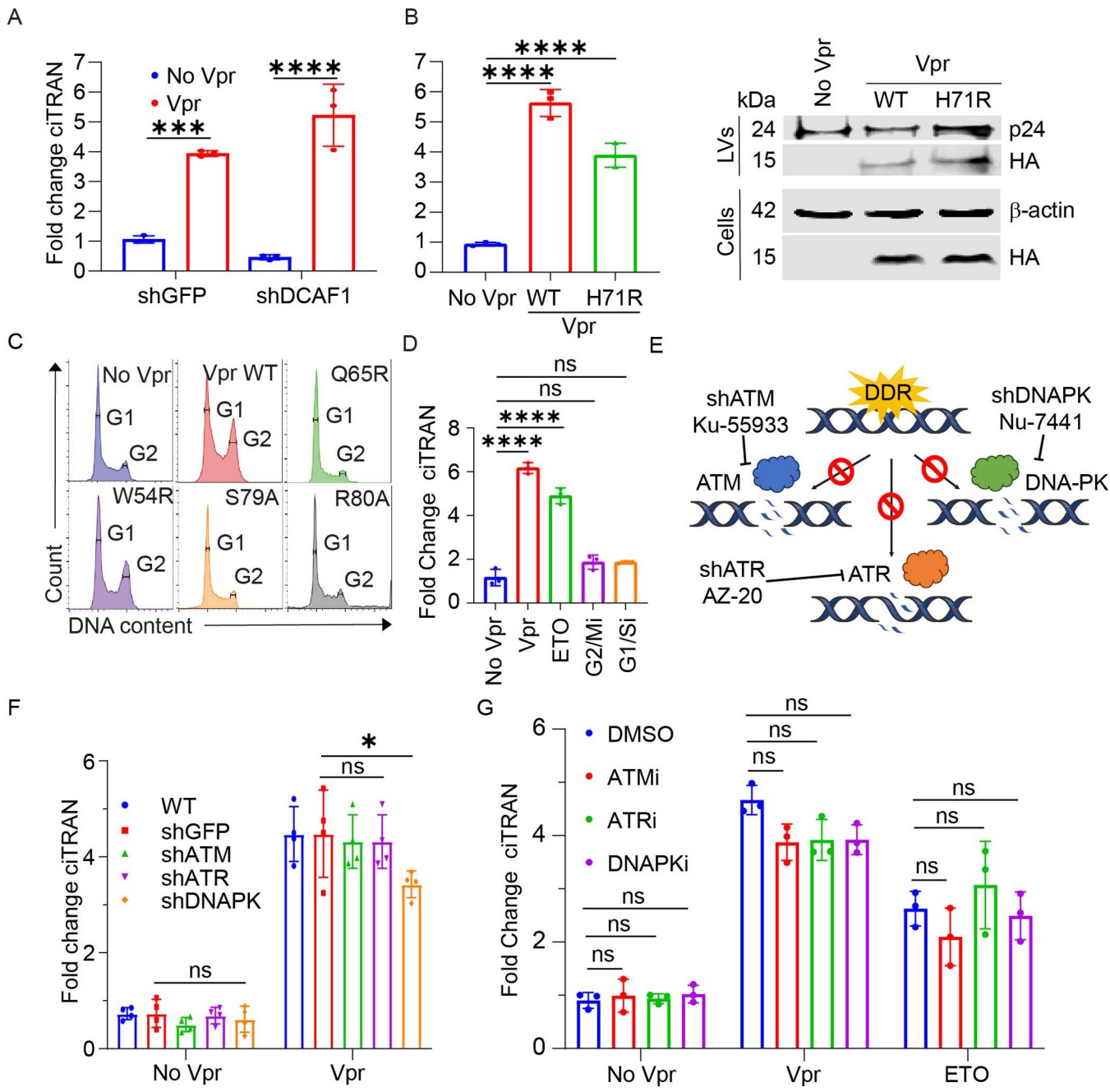

**Fig 2. ciTRAN upregulation independent of DCAF1 or central players of DDR or cell cycle arrest.** (A) ciTRAN levels from JTAg cells expressing stable shGFP or shDCAF1 challenged with LVs produced by transfecting Vpr expressor or empty vector and quantification of ciTRAN by qRT-PCR after 30h, *GAPDH* served as control for normalization. (B) ciTRAN levels by qRT-PCR after 30h from JTAg cells transduced with LVs bearing WT, H71R Vpr proteins, *GAPDH* served as control for normalization (left panel). Immunoblot showing HA-Vpr WT and HA-Vpr H71R from the LV particles lysates and corresponding producer HEK293T cell lysates. P24 and Actin served as respective loading controls (right panel). (C) Cell cycle profiles of JTAg cells at 48h after the challenge of LV bearing indicated Vpr proteins. (D) ciTRAN levels from JTAg cells challenged with LVs bearing Vpr as in (B) or treatment for 6hrs with 10µM ETO or for 24 hours with 10µM cell cycle inhibitors Apigenin or CPI203. Cells were harvested after 24 hours for quantification of ciTRAN by qRT-PCR, *GAPDH* served as control for normalization. Central players of DDR viz ATM/ ATR/DNAPK (E) were knocked down by lentiviral RNAi (F), or inhibited by specific small molecule inhibitors (G), to assess the impact on ciTRAN levels. Vpr containing LVs were used to deliver the protein to JTAg wild type (WT) cells or shGFP or shATR or shATM or shDNAPK stably expressing cells. After 24 hours, the intracellular levels of ciTRAN were assessed by qRT-PCR. The two-tailed Student's t-test (unpaired) or one-way ANOVA with Dunnett's Multiple comparison test was used to assess the significance between two or more groups, ns= non-significant, *p < 0.05, **p < 0.01, ***p < 0.001 and ****p < 0.0001.

[57] alongside to independently confirm the role of DCAF1. In DCAF1-depleted cells, as well as cells that were challenged with LV particles encapsidated with either WT Vpr or H71R mutant, displayed comparable levels of ciTRAN induction (Fig 2A, 2B; 2B right panel), suggesting that DCAF1 depletion or loss of DCAF1 binding of Vpr may not lead to loss of ciTRAN induction potential of Vpr.

HIV-1 Vpr is known to arrest the cell cycle, which is dependent on the presence of its C-terminal tail. Accordingly, truncation of the C-terminal tail or substitution of the 80th arginine residue with alanine prevents its ability to arrest cells in the G2 phase [58]. Our experiments using S79A and R80A mutants suggested that cell cycle arrest may not lead to upregulation of ciTRAN expression (Fig 1B). Furthermore, the W54R mutant was inducing G2 arrest and was promoting DNA damage that upregulates ciTRAN expression (Figs 1B, 1C and 2C). We sought to further dissect this by adding specific inhibitors that arrested cells in particular cell cycle stages (G2/Mi: Apigenin and G1i: CPI203). The ciTRAN levels of the JTAg T cells challenged with the indicated inhibitors, LV-delivered WT Vpr or ETO, were subsequently checked. While the compounds effectively inhibited the cell cycle in the respective phases (S2B Fig), they failed to induce ciTRAN expression (Fig 2D), additionally confirming that cell cycle arrest alone may not result in upregulation of ciTRAN.

Next, because DNA damage precedes ciTRAN upregulation, we asked whether the central players of the DDR pathway, such as ATM, ATR and DNAPK, can mediate ciTRAN upregulation (Fig 2E). Accordingly, JTAg T cells were selected by puromycin after LV transduction for shRNA-mediated knockdown (KD) of ATM, ATR, and DNAPK. A nonrelevant shRNA (to GFP) expressing cell pool was also generated parallelly. These shRNA-expressing cells were subsequently challenged with LV particles encapsidating Vpr. While specific shRNA, consequently reduced the mRNA levels of ATM, ATR and DNAPK (S2C Fig), they had an insignificant effect on the levels of ciTRAN induced by Vpr (Fig 2F). This was independently confirmed by using select inhibitors for each of the central DDR players: Ku-55933 (ATM), AZ-20 (ATR), and Nu-7441 (DNAPK) (Fig 2E). Of note, the ciTRAN induction by ETO as well as Vpr remained unaffected in these conditions (Fig 2G). Taken together, these experiments suggest that cell cycle arrest, the central players of DDR studied or DCAF1 are dispensable for ciTRAN upregulation.

## TGF-SMAD axis orchestrates ciTRAN upregulation

Given that DNA damage correlates with ciTRAN expression, the pathways activated in response to DNA damage could mediate transcriptional induction of ciTRAN (Fig 1E). Notably, the upregulation of ciTRAN as well as its parental mRNA (*SMARCA5*) by WT Vpr as well as other mutants, excluding Q65R, was consistent (S3A Fig), suggesting a promoter level induction of the primary transcript. Emerging evidence indicates that TGF-β pathway gets activated in response to treatment with DNA-damaging agents, such as doxorubicin and etoposide, in cancer patients as well as in cell lines and can affect the clinical outcomes of patients undergoing chemotherapy [39,42]. Moreover, our recent study reported that TGF-β pathway can get activated downstream of sequence-specific DNA cleavage via programmable nucleases such as Cas9 [59]. Strikingly, the treatment of human primary CD4+ cells with ETO (20µM) induced ciTRAN levels (~5-fold) and this was inhibited (~1.5-fold) by the treatment of Repsox, a selective and potent inhibitor of TGF-β receptor 1/ALK5 (S3B Fig). Hence, we speculated that TGF-β pathway activated downstream of Vpr expression may serve as an intermediate signaling module necessary for the upregulation of ciTRAN.

To test this, reporter assays in HEK293T cells were performed using the TGF-β sensitive luciferase plasmid (SBE-Luc; Fig 3A top panel) to determine first whether Vpr can induce TGF-β/SMAD axis. In this construct, the luciferase expression is driven by a promoter carrying SMAD binding element (SBE). Along with SBE-Luc, HEK293T cells were also transfected with either vector or Vpr (WT) or (Q65R) vectors. In addition, cells were transfected with SBE-Luc along with a *gCXCR4*-Cas9-encoding plasmid as control, which generated sequence-specific DNA breaks guided by gRNA specific to *CXCR4* [59]. Recombinant TGF-β served as positive control in these assays and Repsox (10µM) to check the TGF-β dependency. Notably, the luciferase reporter assay indicated that WT Vpr expression, in addition to *gCXCR4*-Cas9 expression, promoted SBE-Luc expression (Fig 3A). However, Vpr (Q65R) or Vpr WT challenged with Repsox failed to

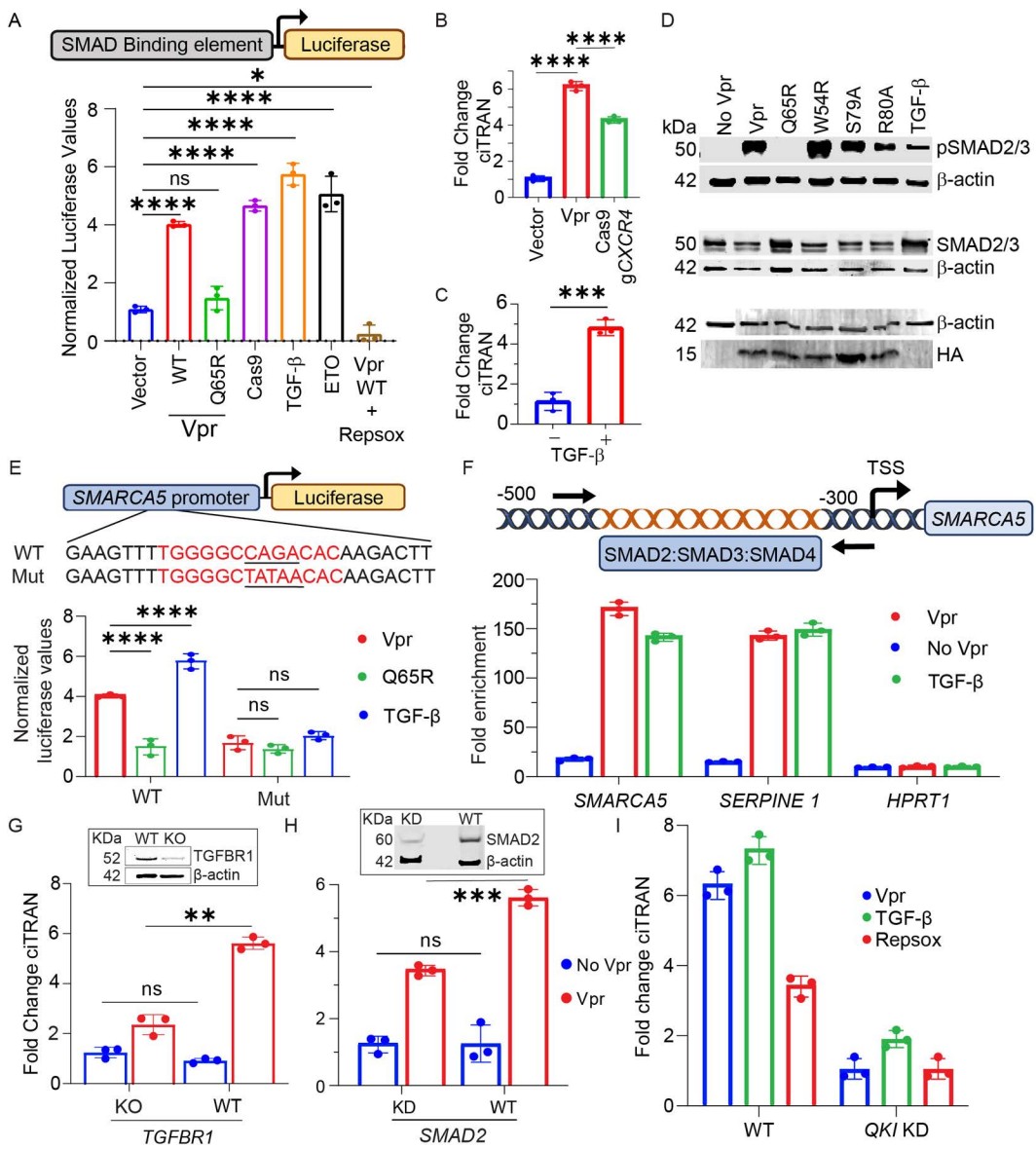

**Fig 3. Vpr triggers TGF-β production orchestrating ciTRAN induction.** (A) The luciferase assay to measure SMAD-dependent luciferase expression from HEK293T cells transfected with SMAD-sensitive luciferase reporter (SBE Luc) along with either empty vector, or plasmids expressing Vpr (with/without Repsox 10µM) or Cas9 or treated with ETO (Etoposide 10µM). Recombinant TGF-β (5ng/ml) was used as a positive control. Data were normalized by total protein amount. ciTRAN levels from: (B) HEK293T cells transfected with empty vector or Vpr or Cas9 expressors, and (C) JTAg cells challenged with TGF-β (10ng/ml), at 24 hours. *GAPDH* serves as control (n = 3 ± SD). (D) Immunoblots from the lysates of JTAg challenged with LVs bearing HA-Vpr proteins from producer HEK293T (bottom panel). Recombinant TGF-β (10ng/ml) treatment served as control. Levels of the p-SMAD2/3 and total SMAD2/3, were checked after 30 hours of transduction. β-actin for each of the target is probed. (E) Top panel: Schematics of putative WT SMAD2/3/4 or Mut-*SMARCA5* promoter regions (details in methods) inserted upstream to the Luciferase gene in pGL3-basic. Bottom panel: Luciferase activity readout after 24 hours from HEK293T cells transfected with WT or Mut *SMARCA5* promoter plasmids along with Vpr expressing or Vpr Q65R expressing vector or empty vector. Recombinant TGF-β1 served as positive control (n = 3 ± SD). (F) Top panel: schematics of putative SMAD binding motifs in *SMARCA5* promoter at the indicated locations. The arrows show the position of oligos (used in ChIP assay) binding to the promoter region with respective positions indicated upstream to the transcription start site (TSS). Bottom panel: Enrichment of SMARCA5 promoter fragment as quantified after transducing Vpr LVs or TGF-β (10ng/ml) via SMAD2/3 specific antibody by ChIP-qPCR assay. IgG served as antibody specificity control (n = 3, technical replicate). *SERPINE1* was used as positive control whereas *HPRT1* intronic region as negative control for SMAD2/3 chromatin recruitment. (G) ciTRAN levels from JTAg cells by qRT-PCR assessed after 30h challenge of LV particles containing Vpr or no Vpr from: (G) Wild-type (WT) or *TGFRI* knockout (KO) cells and (H) control shRNA (SMAD2 WT) or *SMAD2*-specific (*SMAD2* KD) shRNA expressing cells, (n = 3 ± SD). Data were normalized to actin mRNA

levels. Corresponding immunoblots of TGF-β RI and SMAD2 from G and H in the insets with respective loading shown by probing β-actin from the same cell lysates. (I) Effect of QKI knockdown on the upregulation of ciTRAN by Vpr, TGF-β and the sensitivity to Repsox. The two-tailed Student's t-test (unpaired) or one-way ANOVA with Dunnett's Multiple comparison test (A, B, E) or Two-way ANOVA with Sidaks Multiple comparison test (G, H) was used to assess the significance between two or more groups, ns = non-significant, *p < 0.05, **p < 0.01, ***p < 0.001 and ****p < 0.0001.

induce SBE-luciferase expression, emphasizing the specificity of the assay and a hitherto unknown ability of WT Vpr to trigger TGF-β pathway (Fig 3A). In agreement, g*CXCR4*-Cas9-induced sequence-specific DNA cleavage (Fig 3B) or the addition of recombinant TGF-β (Fig 3C) in HEK 293T upregulated ciTRAN levels more than 4-fold. Additionally, in JTAg T cells, the TGF-β signaling effectors SMAD2/3 were also examined for activating phosphorylation, and in agreement, we found that cells transduced with LV particles containing WT Vpr, and to a varying extent other Vpr mutants, except Q65R, presented elevated levels of p-SMAD2/3 as examined by western blotting (Fig 3D).

Intriguingly, analysis of the promoter sequence of *SMARCA5,* the parental gene that encodes ciTRAN, revealed the presence of a SMAD binding consensus motif, which suggested plausible recruitment of crucial TFG-β signaling effectors (SMAD2/3) to promote circRNA expression. To validate this *in silico* observation, we selected a promoter fragment (1157 bp) of *SMARCA5* and cloned it upstream of the luciferase gene. Notably, we found *SMARCA5* promoter-driven luciferase expression was dependent on either the expression of Vpr or the challenge of cells with recombinant TGF-β (Fig 3E). Strikingly, the Vpr Q65R mutant or the promoter variant Mut-*SMARCA5* (which lacked the predicted SMAD binding element) could not induce luciferase reporter expression under these conditions (Fig 3E), suggesting the dependency on the TGF-β canonical signaling activation and the presence of a consensus SMAD-binding element in the promoter for reporter expression. To further independently confirm the *SMARCA5* promoter occupancy by SMAD2/3 and to rule out indirect effects on luciferase expression from the transfected plasmid, we examined the endogenous *SMARAC5* promoter (Fig 3F, top panel) in chromatin immunoprecipitation (ChIP) assays using SMAD2/3 antibody. Compared with the Vpr minus condition, the antibody enriched the *SMARCA5* promoter sequence 169-fold from the lysates of JTAg cells (Fig 3F) challenged with WT Vpr-loaded LV particles. Recombinant TGF-β was used as an inducer of TGF-β pathway. This altogether indicated that Vpr-induced TGF-β can promote SMARCA5/ciTRAN expression via SMAD2/3 recruitment on the *SMARCA5* promoter (Fig 3F, bottom panel).

Additionally, the specificity of the assay was also confirmed by selecting a known downstream effector gene, *SERPINE1*, as a positive control and *HPRT* (intron) as a negative control (Fig 3F, bottom panel). In conditions where TGF-β pathway was stimulated, either by WT Vpr or by direct engagement of TGF receptors by the recombinant ligand, we found that the chromatin region of *SERPINE* but not that of *HPRT* was enriched.

Next, we investigated upstream regulators of TGF-β pathway, starting with CRISPR targeting of *TGFBRI* to determine whether the major receptor is required for ciTRAN upregulation by Vpr. Indeed, genetic ablation of the *TGFBR1* locus affected the ability of Vpr to upregulate ciTRAN; in the gene-edited bulk population, the expression difference was reduced to 1.5-fold from 6-fold (Fig 3G). Western blotting correspondingly confirmed the loss of the TGFRI signal in the gene-edited bulk population (Fig 3G, inset). The degree of TGF dependency was also evaluated by knocking down the primary TGF effector downstream to *TGFBR1* SMAD2 (Fig 3H and inset) and alternatively by pharmacologically targeting TGFBR1 by Repsox (S3C Fig). As expected, we observed a reduction in the magnitude of ciTRAN induction by Vpr in SMAD2-knockdown cells as well as in Repsox-treated cells. However, importantly, regardless of TGF-β pathway activation by Vpr or recombinant TGF-β, ciTRAN upregulation required the QKI protein (Figs 3I and S3D). Taken together, these experiments provided new insights into the positive regulation of TGF-β pathway by HIV-1 Vpr and the concomitant, Repsox-sensitive, ciTRAN upregulation.

## TGF-β receptor 1 blockade inhibits HIV-1 proviral transcription

We next sought detailed insights into the regulation of ciTRAN expression by TGF-β and to determine the contribution of upregulated ciTRAN to the formation of functional RNA polymerase-II (RNAPII) transcription complex on HIV-1

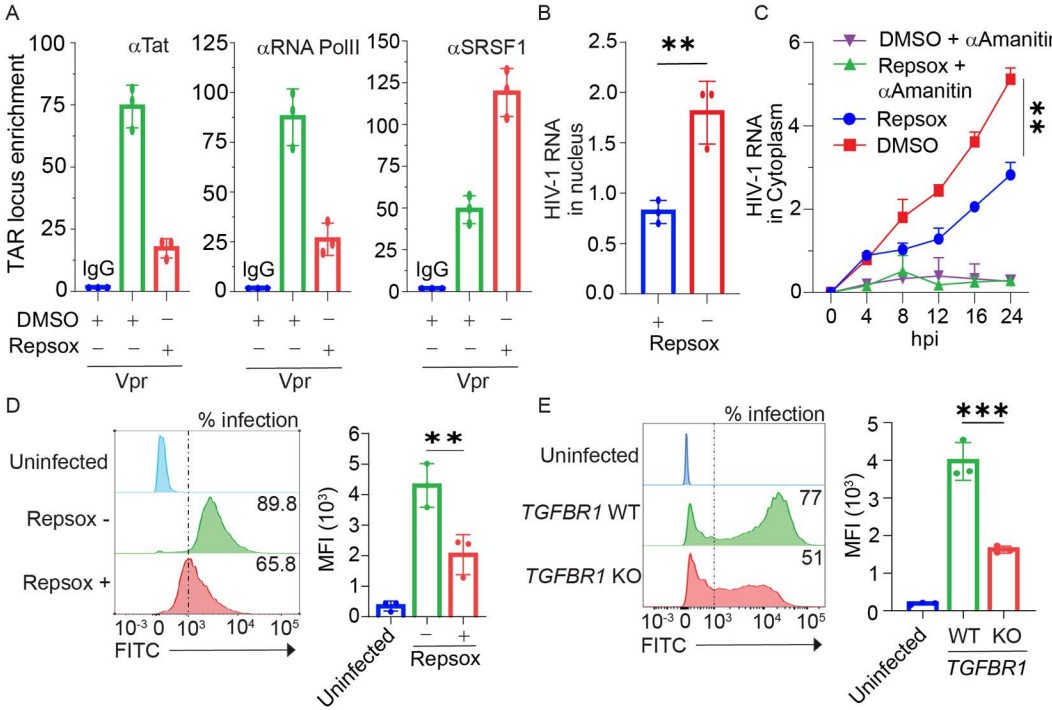

**Fig 4. TGF-β receptor targeting restricts provirus transcription.** (A) TAR locus enrichment and analysis by ChIP-qPCR assay. JTAg cells were challenged with lentiviral particles encapsidating Vpr and were treated with DMSO or Repsox (10μM). The ChIP was performed after 30 hours using RNA PolII-, SRSF1-, and Tat-specific antibody. The data were normalized to TAR region enriched in IgG precipitated fractions (n = 3, technical replicate). Transcription from HIV-1 provirus analysed by gag-specific qRT-PCR in Repsox treated and untreated conditions: (B) Nuclear run-on assay to measure the levels from infected cell nucleus and (C) cytoplasmic fraction collected across indicated time points to measure gag RNA (HIV-1 RNA) levels from the cytoplasm (α-Amanitin served as a control). The data were normalized in each condition to *GAPDH* mRNA. (D, E) The frequency (expressed as percent positive cells) and the magnitude of HIV-1 gene expression (expressed as Mean Fluorescence Intensity of HIV-1 encoded zsGreen reporter (MFI) captured by flow-cytometry after 24 hours of infection of E6.1 T cells treated +/- Repsox (D), and JTAg T cells lacking functional *TGFBR1* (E), (n = 3; ± SD). The two-tailed Student's t-test (unpaired) or one-way ANOVA with Dunnett's Multiple comparison test. was used to assess the significance between two or more groups, ns = non-significant, *p < 0.05, **p < 0.01, ***p < 0.001 and ****p < 0.0001.

LTR promoter. Accordingly, we examined SRSF1, RNAPII, and Tat occupancy via ChIP assays in conditions where Repsox or DMSO inhibited TGF-β pathway. A strong interaction of SRSF1 with the HIV-1 TAR locus was observed in conditions where the cells were challenged with Repsox (Fig 4A), which is consistent with our previous study [17] in a manner that recapitulates a competing scenario between ciTRAN and SRSF1 in influencing HIV-1 proviral transcription (Fig 4A). These findings were also consistent with the reduced association of RNAPII and Tat with the TAR region in Repsox-treated cells (Fig 4A). TGF-β signaling from the receptor inhibited by Repsox concurrently blunted ciTRAN expression (S3C Fig), which negatively impacted the steady-state levels of provirus-encoded HIV RNA as examined from nuclear and the cytoplasmic compartments (Fig 4B and 4C). Correspondingly, decreased replication of HIV-1 in the presence of Repsox in E6.1 T cells was also observed (Fig 4D). Overall, the levels of viral transcripts and protein expression were impacted by Repsox treatment, as reflected by the frequency of reporter (zsGreen) positive cells and the magnitude of expression detected by flow cytometry (Fig 4D; 4D right panel). Moreover, a CRISPR assay with the *TGFBRI* primary receptor in JTAg T cells and subsequent infection by a zsGreen-expressing HIV-1 was performed to confirm this phenomenon independently (Fig 4E), altogether confirming the positive impact of TGF-β induction on transcription from the provirus.

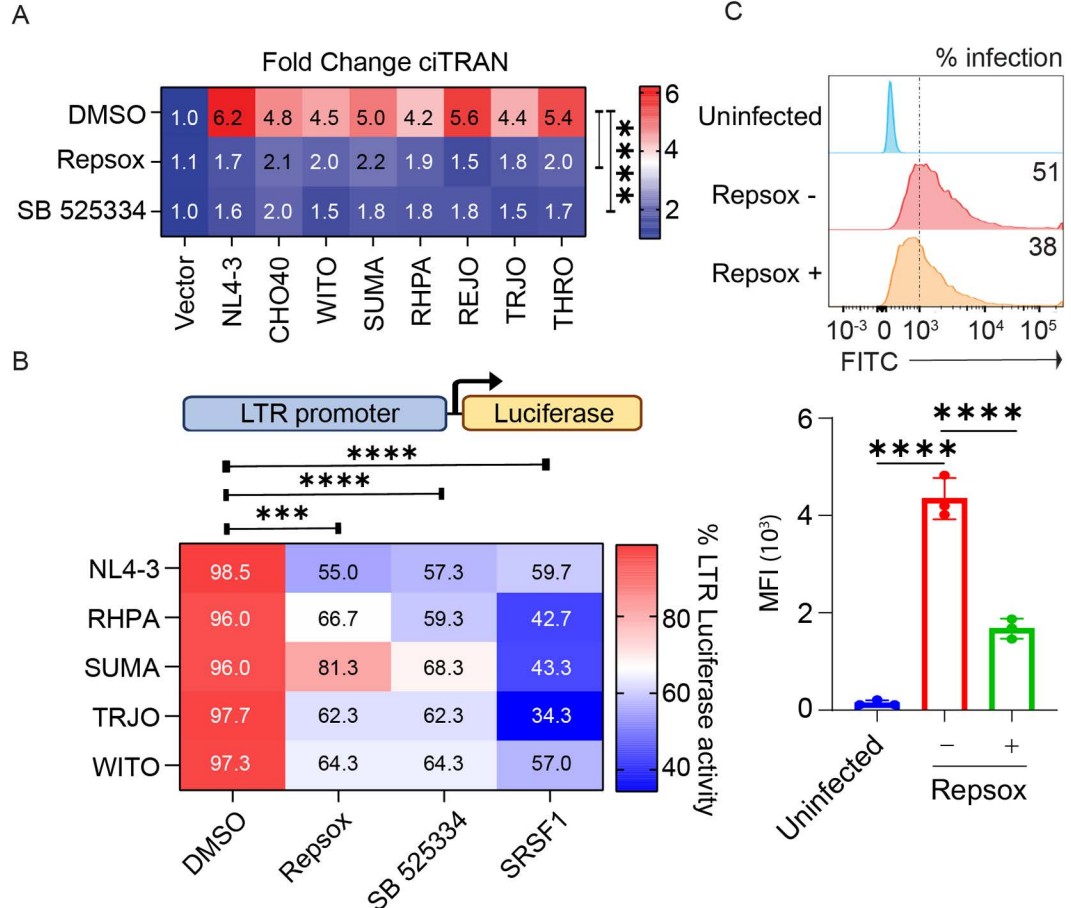

**Fig 5. Effect of TGF-β pathway inhibitors on TF virus-mediated ciTRAN upregulation and transcription from LTRs.** (A) Relative ciTRAN levels (quantified by qRT-PCR) from indicative TF virus molecular clone transfected HEK293T cells that were treated either with TGF-β inhibitors Repsox (10μM) or SB525334 (10μM) or DMSO. (B) Transcription from LTRs of indicated TF viruses (cloned upstream to the luciferase in PGL3 basic vector) and its sensitivity to TGF-β inhibitors Repsox (10μM) or SB525334 (10μM) or DMSO, as assessed by luciferase activity assay in HEK293T cells. Data were normalized to total protein quantified by Bradford assay. SRSF1 ectopic expression was used as positive control for the assay performed. (C) The frequency (expressed as percent positive cells) and the magnitude of HIV-1 gene expression expressed as Mean Fluorescence Intensity of HIV-1 encoded zsGreen reporter (MFI)) captured by flow-cytometry after 24 hours of infection of THP-1 treated +/- Repsox. The two-tailed Student's t-test (unpaired) or one-way ANOVA or two-way ANOVA with Dunnett's Multiple comparison test was used to assess the significance between two or more groups, ns = non-significant, *p < 0.05, **p < 0.01, ***p < 0.001 and ****p < 0.0001.

### TGF-β pathway blockade suppresses transcription from LTRs of transmitted founder (TF) viruses

Whether small-molecule inhibitors of TGF-β pathway can reduce the ciTRAN levels induced by TF viruses was checked next. Accordingly, two distinct small molecules (Repsox and SB525334) known to target TGF-β pathway were used to challenge HEK293T cells transfected with molecular clones of the indicated TF viruses, and ciTRAN levels were captured subsequently by qRT-PCR. We observed decreased ciTRAN levels in cells challenged with both inhibitors (Fig 5A). Furthermore, the effect of small molecules was also tested on luciferase driven by LTRs of the indicated TF viruses as well as NL4–3-derived LTR, and reduced transcription was observed for all the LTRs tested (Fig 5B, SRSF ectopic expression was used as a control in these reporter assays as reported earlier [17]), suggesting the conserved nature of these signaling cascades in the replication of clinically relevant isolates. Moreover, we also investigated whether virus replication in monocytic lineages (such as THP-1), where Vpr also impacts viral gene expression [31], is sensitive to Repsox treatment.

Indeed, Repsox treatment decreased the expression and frequency of zsGreen-positive THP-1 cells (zsGreen encoded by the provirus as a marker of viral gene expression) (Fig 5C; 5C bottom panel).

## Repsox inhibits Vpr-driven ciTRAN upregulation in primary cells

We next sought to understand the importance of Vpr-driven TGF-β production on ciTRAN upregulation and for HIV-1 gene expression using the primary CD4+ T cells enriched to purity by positive selection (S4A Fig) from pooled PBMCs of three healthy donors (S1 Table). We first checked if Vpr can induce the major TGFB isoforms that differ in sequence identity (S4B Fig). The expression of the isoforms was checked upon the Vpr LV particle challenge of primary CD4+ T cells. TGFB1 mRNA~3 fold, and to a lesser extent TGFB2 mRNA~2 fold, was upregulated in Vpr-expressing CD4+ primary cells (S4C Fig). Moreover, ciTRAN expression was elevated either by treatment with recombinant TGF-β1 or by infection of primary CD4+ T cells by NLBN-zsGreen reporter virus encoding Vpr and was reversed by treatment with Repsox (Fig 6A). Notably, reduced ciTRAN levels upon challenge by Repsox were not associated with reduced viability of primary CD4+ cells in these conditions (S4D Fig). ELISA from the cell-free supernatant (from Fig 6A) confirmed that infection with NLBN zsGreen reporter virus strikingly induced TGF-β1 expression (380pg/ml) and that it was inhibited by the addition of Repsox to 39 pg/ml (Fig 6B). Specifically, in this case, the advanced RPMI was used to grow the cells containing minimal FBS (1%) to exclude the possibility of TGF-β1 being non-specifically detected in ELISA from the bovine source. In agreement, the impact on the proportion of positive cells, as well as the magnitude of zsGreen reporter expression, was negatively influenced by Repsox treatment (Fig 6C). Repsox reduced reporter-positive cells from 66% to 45% and reporter MFI from 4.2-fold to 1.4-fold, confirming Repsox suppresses HIV-1 gene expression in primary cells in these experimental settings.

Moreover, from PHA/IL2 stimulated pool of PBMCs from three donors, we performed ChIP using SMAD2/3 specific antibody to see if the induction of ciTRAN is due to recruitment of SMAD2/3 on the chromatin as done at 30h in Fig 3F. SMAD2/3 was enriched (~4 fold) in Vpr LV particle challenge cells in comparison to the mock. Interestingly, the transcription factors were retained at an early time point (12h) on the SMARCA5 promoter but dissociated at a later time point (30h) in recombinant TGF-β treated cells suggesting the requirement of a constant stimulus by purified TGF-β ligand for SMAD2/3 chromatin association (Fig 6D,6E). Additionally, from CD4 cells isolated from three different healthy donors, we also assessed the levels of ciTRAN by qRT-PCR and cell-free TGF-β1 by ELISA. These CD4+ cells were challenged with LV particles +/- Vpr protein and the challenged cells additionally received Repsox. The ciTRAN levels ranged between ~2–6 fold while the cell-free TGF-β was between ~100–210 pg/ml and were Repsox sensitive (Fig 6F, 6G). These experiments indicated that Vpr increases ciTRAN expression in primary cells, and this increase is associated with higher levels of TGF-β.

To understand the relevance of TGF-β for HIV-1 replication, next we captured the replication kinetics of HIV-1 NL4–3 (R+) and a Vpr-defective counterpart NL4–3 (R-) from PHA/IL2 stimulated PBMCs from three pooled donors. The stimulated PBMCs were subsequently infected for 6 hrs with the indicated virus (R+/R-) and, post-infection, kept under Repsox (10μM) or vehicle (DMSO) from day zero onwards. At indicated time points, the cells were collected for analysis of intracellular p24 expression by flow cytometry (Fig 6H). We found that while R+ and R- virus replicated with no difference on day 1 and a marginal difference on subsequent days, the R+ virus in the presence of Repsox consistently replicated with poor kinetics (p24 percent positivity on day 6 at peak infection: 43.3±2% to 24.3±0.57%). We also assessed the magnitude of virus gene expression by capturing MFI from day 3 onwards (Fig 6I). We found that the magnitude of expression of p24 of R+ was impacted by the Repsox treatment plausibly suggesting the impact on gene expression mediated by TGF-β in these experimental conditions. Furthermore, from the RNA isolated from these cells collected at peak infection on day 6, we found corresponding upregulation of ciTRAN by qRT-PCR and its sensitivity to Repsox (Fig 6J). ELISA from the cell-free supernatant from this timepoint indicated that TGF-β levels were only affected in case of the R+ virus infected condition by Repsox from ~308 pg/ml to 218 pg/ml (Fig 6K). The cell-free TGF-β was also detected from R- virus infection

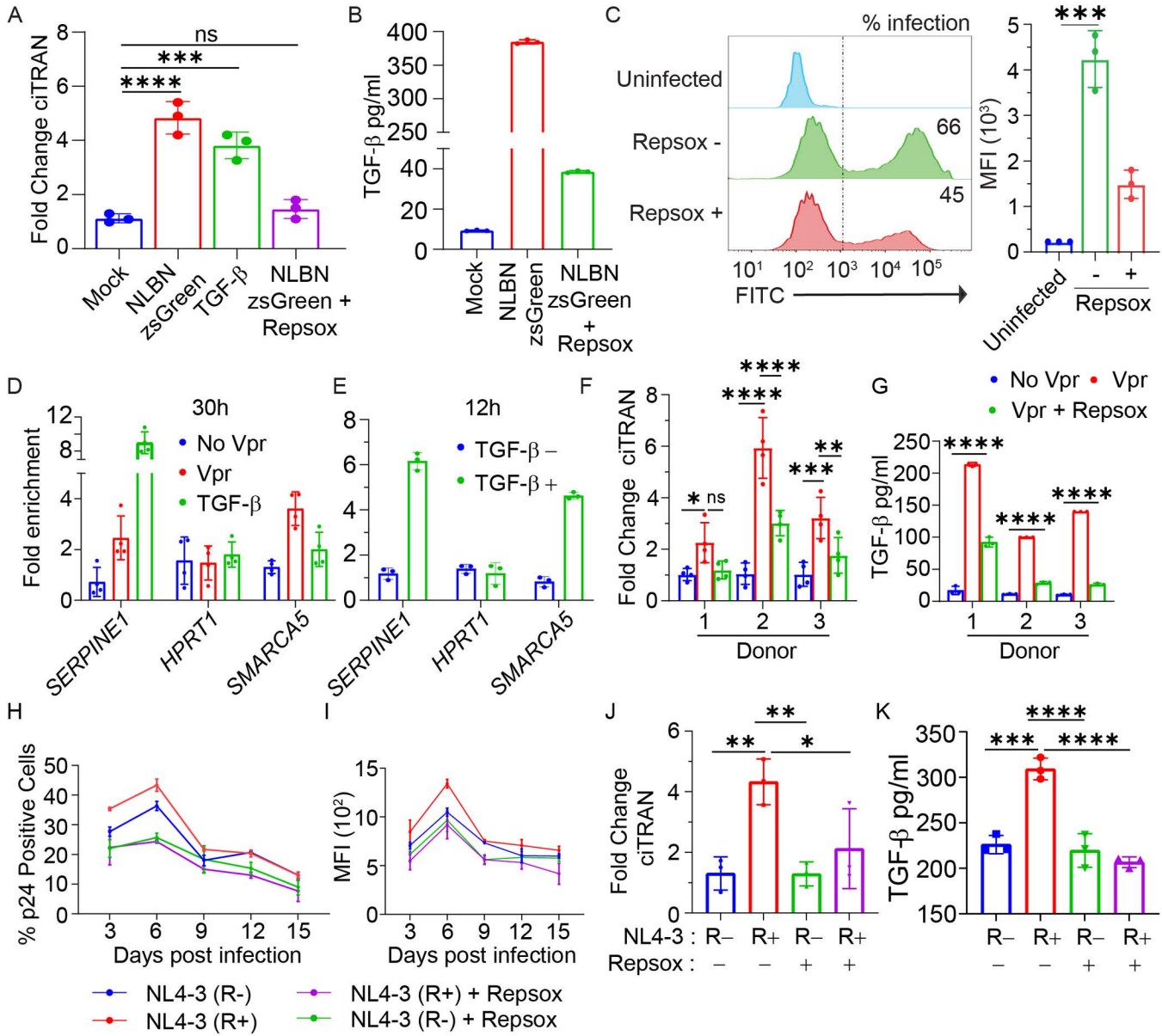

**Fig 6. ciTRAN expression in primary cells upregulated by Vpr and sensitivity to Repsox.** (A) ciTRAN level analysed using qRT-PCR from CD4+ cells after infecting them with HIV-1 NLBN zsGreen/ Repsox (+/-) or treatment with recombinant TGF-β (10ng/ml) (n=3; ±SD). (B) Corresponding supernatants from (A) were assessed for cell-free TGF-β by ELISA (n=3, technical replicates). (C) Effect of Repsox on HIV-1 NLBN zsGreen replication in primary CD4+ cells after 30 hours of infection. MFI, mean fluorescence intensity, (n=3; ±SD). The frequency (expressed as percent positive cells) and the magnitude of HIV-1 gene expression expressed as Mean Fluorescence Intensity of HIV-1 encoded zsGreen reporter (MFI) captured by flow cytometry. (D) The enrichment of the *SMARCA5* promoter fragment from PHA/IL2 stimulated PBMCs, at 30h (D) and 12h (E) after challenge with Vpr-carrying or Vpr-null LVs or TGF-β (10 ng/ml), by ChIP-qPCR assay with a SMAD2/3-specific antibody. IgG served as an antibody specificity control. *SERPINE1* was used as positive control whereas *HPRT1* intronic region was taken as negative control for SMAD2/3 recruitment (n=3, technical replicates). (F) ciTRAN levels were checked from primary CD4+ T cells from three donors after transducing with Vpr bearing LVs with or without Repsox after 30 hours and (G) corresponding supernatants were subjected to TGF-β1 ELISA (n=3, technical replicates). (H) Intracellular p24 staining on days post-infection from PHA/IL2 stimulated PBMCs that were challenged with HIV-1 NL4–3 (R+) and NL4–3 (R-) viruses and cultured with or without Repsox (10 μM). (I) Corresponding mean fluorescence intensity (MFI) values of immunostained p24 indicative of the magnitude of viral gene expression. (J) ciTRAN levels quantified by qRT-PCR, and (K) TGF-β ELISA from supernatants were determined from samples collected at day 6. The two-tailed Student's t-test (unpaired) or one-way ANOVA with Dunnett's Multiple comparison test was used to assess the significance between two or more groups, ns=non-significant, *p<0.05, **p<0.01, ***p<0.001 and ****p<0.0001.

(~205 pg/ml), suggesting infection-associated cytokine pool, but it was insensitive to Repsox treatment. Altogether, WT HIV-1 replication was negatively influenced by Repsox treatment (Fig 6H, 6I), suggesting that Vpr-induced TGF-β may promote WT virus replication in primary cells.

To further investigate the role of exogenous TGF-β in HIV-1 replication we additionally conducted replication assays using HIV-1 NL4–3 (R+) and its Vpr-deficient counterpart (R-) in PHA/IL-2-stimulated PBMCs from three independent donors. Following 6-hour infection, cells were cultured in the presence of Repsox (10 µM), recombinant TGF-β (10 ng/mL), or vehicle (DMSO), and intracellular p24 expression was assessed by flow cytometry on day 3.

Interestingly, recombinant TGF-β failed to rescue the replication defect of the R- virus, indicating that ciTRAN induction alone may be insufficient and that a coordinated action of both Vpr and ciTRAN may be required for full transcriptional activation in these experimental conditions. Moreover, Repsox treatment led to similar inhibition in both R+ and R- viruses in these experimental conditions (S5A Fig), which may also account for donor specific effects. While intracellular p24 staining is a reliable indicator of viral protein expression at the cell population level, the inherent stability of p24 may additionally confound dynamic assessments of viral gene expression events. Therefore, a comprehensive understanding of replication dynamics is further needed in order to get detailed insights. Independently, we further assessed LTR-driven transcriptional activity using NL4–3 Env − R − Luc and NL4–3 Env − R + Luc reporter viruses. For this, CD4 + T cells were infected at an MOI of 5, and luciferase activity was measured 24 hours post-infection across the indicated experimental conditions. The luciferase activity data indicated that Repsox, in addition to other TGF-β pathway inhibitors, significantly inhibited LTR-driven reporter for R+ virus, but had insignificant effect in the absence of Vpr (S5B Fig), altogether suggesting that TGF-β alone may be insufficient to compensate for the absence of Vpr.

## Discussion

We demonstrate that HIV-1 Vpr stimulates the TGF-β production, which in turn mediates the upregulation of ciTRAN. Vpr is a unique protein encoded by primate lentiviruses known to promote viral replication [18,32], and the selective pressure exerted to maintain the functional copy of a gene implicates its relevance for natural infection [34,60,61]. Notably, the glutamine residue at the sixty-fifth position is remarkably conserved across the HIV-1 Vpr sequences we analyzed (S4E Fig), implying a selective pressure exerted to boost the TGF-β production.

SRSF1, a protein known to impact viral gene expression significantly [62], was shown to be antagonized by Vpr by promoting the expression of ciTRAN [17]. However, the molecular mechanism of ciTRAN induction by Vpr remained elusive. The induction of ciTRAN by HIV-1 Vpr appears to be actively achieved by the cascade of events mediated by the accessory protein. The experiments from cell lines of distinct tissue origins and primary CD4+ cells revealed that Vpr-induced DNA damage precedes the induction of TGF-β pathway. The TGF-SMAD axis subsequently orchestrates ciTRAN upregulation in these experimental conditions. This is because targeting the receptor for TGF-β by orthogonal means blunted ciTRAN expression during Vpr-proficient virus infection. Interestingly, ciTRAN expression is also induced, independent of infection, by DNA damaging agents like ETO/Dox or by the addition of recombinant TGF-β bypassing the DNA damage. Considering the TGF-β pathway can also influence the DNA repair outcomes [39,40], there is a possibility that ciTRAN may be involved in DNA repair outcomes in T cells.

The experiments also demonstrate how circRNA expression and its proviral function in this context can be inhibited by a small molecule Repsox. Repsox potently and selectively inhibits TGFBR1/ALK5 actions by suppressing the binding of ATP to the receptor and its subsequent phosphorylation to inhibit the signaling with IC50 of 4 nM [63,64]. Repsox was found to reduce the frequency of infected cells and magnitude of viral antigen expression. Chronic infections like HIV-1 are associated with enhanced TGF-β signaling in T cells and diminished T cell responses. Conversely, TGF-β deficiency leads to autoimmunity, sterility, and shortened lifespan in mice [65–67]. While the TGF-β pathway regulates T cell differentiation and homeostasis [68], the DNA damage–induced TGF-β response in T cells may be distinct, and its role in HIV-1 replication, particularly in the context of Vpr, remains to be explored in detail. Notably, while our data rule out canonical

DDR factors (ATM, ATR, DNA-PK), they do not address the interplay between DNA damage during infection [69] and Vpr-induced TGF-β production. HIV-1 appears to utilize discrete strategies to modulate DDR signaling and antiviral defense mechanisms; Vpr along with Vpu and Vif have also been shown to modulate DDR pathways [70,71]. Additional possibilities include the stimulation of the oxidative stress pathway by the primate lentiviral protein, which precedes the induction of TGF-β, warranting additional investigation. [72,73]. Furthermore, whether IL6 is involved in an additional positive feedback mechanism [74] remains an intriguing possibility. Vpr also targets IL6 and Tet2, which are positive and negative regulators, respectively, of HIV replication [75,76], and how this dynamic interplay influences TGF-β pathway to promote HIV-1 transcription remains to be explored. Regardless, the importance of studying TGF-β induction by HIV-1 Vpr is also stressed by the fact that increased TGF-β signaling can lead to pathologic concentrations of IL6 [77] and the development of Kaposi sarcoma [78].

The study presented herein has been focused on unravelling the mechanism through which HIV-1 Vpr induces ciTRAN, with our findings underscoring the role of Vpr-induced TGF-β in this context. It has a few limitations. Firstly, the action of Repsox as the inhibitor of TGF-β pathway in primary cells infected with HIV-1 will require more investigation to elucidate possible variations in Repsox's effects on viral replication. Secondly, we have not ascertained if Repsox treatment results in elevation of other inflammatory genes in the presence of Vpr, given TGF-β as the pleiotropic cytokine with immunosuppressive properties. Additionally, as the ELISA measured total TGF-β, variations in active TGF-β levels may affect interpretation of Repsox's impact on viral replication, warranting further investigation. We thus acknowledge the complexity of deciphering how TGF-β fine-tunes the host cellular environment during HIV-1 infection, given its pleiotropic role in immune regulation, cell growth, and differentiation. Previous studies, including [43], have shown that infection with Vpr-deficient (R−) virus can also induce TGF-β and enhance HIV-1 infection in both activated and resting memory CD4 + T cells. In our study, we specifically addressed this by using the TGF-β inhibitor Repsox and found that its effects on HIV-1 replication—whether wild-type or Vpr-deficient—do not clearly distinguish between cell-intrinsic (infection-induced) and exogenously supplied TGF-β. Notably, while Vpr-driven TGF-β levels were reversed by Repsox addition, infection-induced TGF-β levels remain unaffected, suggesting that Vpr engages additional, yet unidentified, mechanism to modulate TGF-β signaling.

In sum, while uncovering circRNA upregulation by TGFβ, this study highlights the necessity for a clearer delineation of the protective and pathogenic capacity of TGF-β superfamily cytokines to enable appropriate modulation for therapeutic purposes.

## Methods

The source of various plasmids, cells and reagents used in this study can be found listed in the S1 Table. Oligos sequences and full-blown blots are provided in the S1 File. The raw data values are provided in S1 Data.

**Ethics statement**: The Institute Ethics Committee of Indian Institute of Science Education and Research Bhopal approved the study (IISERB/IEC/Certificate/2018-II/04). Formal written consent was obtained from the donors, and anonymity was maintained. For additional validation experiments, PBMCs were sourced from HiMedia.

## Cell culture

Jurkat E6.1 (ATCC), THP-1 (ATCC), and Jurkat TAg (JTAg) [79] cell lines were cultured in RPMI 1640 (Gibco, catalog no. 11875093) supplemented with 10% fetal bovine serum (Certified, heat-inactivated serum from Gibco) or Advanced RPMI (Gibco, catalog no. 12633–012) supplemented with 2% FBS. HEK293T (ECACC),TZM-GFP [79] cells were maintained in 10% FBS containing Dulbecco's modified Eagle medium (Gibco, catalog no. 12100046) or 2% FBS containing Advanced DMEM (Gibco, catalog no. 12491–015) with 2mM L-Glutamine. The cells were kept in a humidified 5% $CO_2$ incubator at 37°C.

## Lentiviral Vector (LV) particle production

Lentiviral Vector particles encapsidating Vpr proteins were produced by transfecting pScalps ZsGreen (4µg), psPax2 (3 µg), pMD2.G (1µg) along with 2 µg Vpr WT or other mutants expressing vectors (pCDNA-HA-Vprs) or vector alone from HEK293T cells using calcium phosphate transfection. After 12–15h transfection, the medium was replaced with 2% FBS containing DMEM. The virus-containing supernatant collected after 48h was clarified by centrifugation at 500 x g for 5min and filtered using a 0.2 µm Syringe filter. To assess the incorporation of HA-Vpr and mutants into the lentiviral particles by western blotting, the filtered supernatant was layered on a 20% sucrose cushion and ultracentrifuged at 100,000 x g for 2 hours at 4°C. Following ultracentrifugation, the supernatant was aspirated, and the LV pellet was resuspended in Laemmli buffer containing 50 mM Tris(2-carboxyethyl) phosphine hydrochloride (TCEP).

## Virus production and infection

For infection, the virus was produced from HEK293T producer cells using calcium phosphate method by co-transfecting 7 µg NLBN zsGreen and 1 µg pMD2.G and was limited to single-cycle replication [80]. NLBN zsGreen was a kind gift from Prof. Massimo Pizzato, which is a derivative of NL4–3, where the env region is deleted, and the zsGreen fluorescent protein coding gene is cloned in the nef region. The virus-containing culture supernatant was collected 48h post-transfection, centrifuged at 300xg for 5min, and filtered using a syringe filter of 0.22µm. The virus was quantified using an SGPERT assay [81], and the multiplicity of infection (MOI) was calculated by infecting HEK293T cells or TZM-GFP cells. JTAg cells were infected at MOI = 5 or 10. Primary CD4 + T cells were infected at MOI = 5 or 10. For transducing CD4$^+$ cells from three independent donors with Vpr carrying LVs at an MOI of 5, cells were transduced by spinoculation at 1200xg for 2 hours at RT.

**SYBR green I-based product-enhanced reverse transcriptase (SGPERT) assay.** Following the collection and filtration of the virus-containing media, 5 µL of the virus was lysed with 5 µL of lysis buffer containing 0.25% Triton X-100, 50 mM KCl, 100 mM Tris-HCl (pH 7.4), and 40% glycerol. The reaction was incubated for 10 min at room temperature and then mixed with 90 µL of 1 × core buffer, which consisted of 5 mM [NH4]2SO4, 20 mM KCl, and 20 mM Tris-HCl at a pH of 8.3. Further,10 µL of the mixture was combined with 10 µL of 2 × reaction buffer containing 5 mM [NH4]2SO4, 20 mM KCl, 20 mM Tris-Cl (pH 8.3), 10 mM MgCl2, 0.2 mg/mL bovine serum albumin (BSA), 1/10,000 SYBR green I, 400 µM dNTPs, 1 µM forward primer(5′-TCCTGCTCAACTTCCTGTCGAG-3′), 1 µM reverse primer (5′-CACAGGTCAAACCTCCTAGGAATG-3), and 7 pmol/mL MS2 RNA to quantify RT units by qPCR analysis [81].

## RNA isolation, cDNA and qRT-PCR

For RNA isolation, cells were lysed using TRIzol (Invitrogen) or TRI-reagent (Sigma) or QIAzol (Qiagen) and phase-separated using chloroform. Samples were treated using DNaseI (Thermo) to remove DNA contamination. cDNA first-strand was synthesized by random hexamers or Oligo(dT) primers using Revert Aid (H minus). SYBR green-I-based qPCR was performed for expression quantification, and data were represented as relative fold change.

## Ribonuclease R treatment

For Validation of ciTRAN (circ-SMARCA5/ciTRAN) we treated the RNA pool with RNaseR. One microgram of RNA was incubated with RNaseR (1 unit) at 37°C for 30min. RNaseR was then inactivated at 90°C for 10 minutes. RNA was then eluted using a Zymo clean and concentrator kit for qRT-PCR to quantify respective RNAs.

## Alkaline comet assay

A modified alkaline comet assay procedure was followed as previously described [82]. Briefly, JTAg cells were resuspended at 4 × 10⁴ cells/mL in 1 × PBS after infection with Vpr mutants packaged Lentiviral particles. Cells were mixed with

low melting agarose (1%) (VWR Life Science) at a 1:5 ratio and spread over the slide pre-coated with 1% normal agarose. Slides were dried at room temperature for 10 min and immersed into alkaline lysis buffer (10 mM Tris-HCl, pH 10, 2.5 M NaCl, 0.1 M EDTA, 1% Triton X-100) for 4 h and then in the alkaline running buffer (0.3 M NaOH, 1mM EDTA) for 30 min and finally electrophoresed at 300 mA for 30 min, all done at 4°C. Samples were washed thrice with double-distilled water (ddH2O) and fixed in 70% ethanol at 4°C for 10 min. Slides were dried for 2 h at room temperature and stained with EtBr solution (2 µg/mL in water). Images were acquired using a Zeiss Apotome fluorescence microscope (Carl Zeiss) using a 20×objective lens, and the tail moments were quantified using the OpenComet plugin [83] in the software ImageJ. For each condition, at least 50 cells were analysed.

## Cell cycle analysis

For cell analysis, JTAg cells were treated with lentiviral particles carrying Vpr WT or mutants or treated with G2/M inhibitor (Apegenin), and G1 inhibitor (CPI 203). Cells were harvested after 48 hours and fixed in ice-cold 70% ethanol for 2 hours at 4°C. Cells were then washed in 1X PBS and stained with a staining solution containing PI (50µg/ml) and RNaseA (50µg/ml) for 30 minutes and proceeded for their DNA content analysis using flow cytometry (FACS BD AriaIII). The fluorescence of 10,000 cells were analysed using FlowJo software.

## SMAD reporter assay

To check the effect of ciTRAN induction upon inhibition of the TGFBRI pathway, we used the SMAD reporter plasmid SBE4-Luc (16495; Addgene). This plasmid contains the luciferase reporter gene under the SMAD binding elements, which are expressed under TGFBRI-mediated signaling. HEK293T cells were seeded in a 24-well plate 24 h before transfection. Next, cells were transfected with SBE4-Luc, along with pcDNA HA-Vprs or Empty Vector by calcium phosphate method, media was replaced after 12–15 hours further. Repsox (10µM) or Etoposide (10µM) treatment was given after replacing the media. The transfected cells were lysed after 24 hours of transfection using 100 µL lysis buffer (1% Triton X-100, 25mM tricine [pH 7.8], 15mM potassium phosphate, 15mM MgSO4, 4mM EGTA, and 1mM DTT) for 20min at room temperature. Luminescence readings were obtained using a Spectramaxi3X plate reader, by mixing 50 µL cell lysate supernatant with 50 µL substrate buffer (lysis buffer with 1mM ATP, 0.2mM D-luciferin). Data were normalized to total protein quantified by Bradford readings of the cell lysate.

## *SMARCA5* promoter constructs and luciferase assay

To Check the TGF-β/SMAD axis affecting SMARCA5 transcription, we generated *SMARCA5* promoter-luciferase construct. For this, human SMARCA5 (transcript, NM_003601) promoter sequence was retrieved from the Eukaryotic Promoter Database [84]. The *SMARCA5* promoter from −1452 bp to − 343 bp upstream of the TSS at +1 was amplified by PCR using Jurkat E.6.1 T cell genomic DNA as template and cloned into a pGL3-Basic expression vector (Promega, E1751). *SMARCA5* promoter fragment was cloned between the SmaI sites. For the *Mut* SMARCA5 generation, SMAD2:SMAD3:SMAD4 binding sites were retrieved from the Jasper Motif database [85], and site-directed mutagenesis was performed to introduce mutations in the putative SMAD-binding site (GCCAGAC to GCTATAAC). Primer details are given in S1 File. Mutations were confirmed using Sanger sequencing. For luciferase assay, HEK293T cells were seeded in a 24-well plate 24 h before transfection. Next, cells were transfected with *SMARCA5* WT promotor-Luc or *SMARCA5* Mut promotor-Luc with pcDNAHA-Vpr WT or pcDNAHAQ65RVpr or recombinant TGF-β treatment as positive control. The transfected cells were first lysed after 24 hours of transfection using 100 µL lysis buffer (1% Triton X-100, 25mM tricine [pH 7.8], 15mM potassium phosphate, 15mM MgSO4, 4mM EGTA, and 1mM DTT) for 20min at room temperature. Luminescence readings were acquired using a Spectramaxi3X plate reader by mixing 50 µL cell lysate supernatant with 50 µL substrate buffer (lysis buffer with 1mM ATP, 0.2mM D-luciferin). Data were normalized to total protein quantified by Bradford assay of the cell lysate.

## Knockdown and knockout cell generation

SMAD2, ATR, ATM, DNA-PK, DCAF1 and QKI knockdown JTAg cells were generated using pLKO.1 mission shRNAs from Sigma. (shRNA sequences are provided in S1 File) Lentiviral particles were produced from HEK293T by cotransfecting respective shRNA encoding plasmids or control shRNA (shGFP) along with psPAX2, a packaging plasmid and pMD2.G, a VSV glycoprotein encoding plasmid by calcium phosphate method. The medium was replaced after 12-15h post-transfection and lentiviral particles were collected after 48h. Lentiviral particles containing media was centrifuged at 500xg and filtered using syringe filter 0.22 µm. JTAg cells were infected using shRNA particles and selected by puromycin for one week. For the generation of *TGFBRI*-knockout cells, JTAg cells were transduced with lentiviral vectors carrying Cas9 with guide RNA against *TGFBRI* (gRNA sequences are provided in S1 File). Furthermore, transduced cells were selected for one week with 1 µg/mL puromycin.

## Subcellular fractionation and estimation of HIV-1 transcription upon Repsox treatment

Subcellular fractionation was performed as described previously [86]. Briefly, for subcellular Fractionation 5 million per condition JTAg cells infected by HIV-1 NLBN zsGreen with/without the Repsox treatment (10µM) were collected at different time points by spinning down for 500 x g at 4°C for 5min. Cells were lysed with Hypotonic lysis buffer (HLB) with RNase inhibitor. Cells were kept on ice for 10 minutes and then centrifuged at 1000xg for 3min at 4°C. The supernatant was carefully transferred to a new tube with RNA precipitation solution for 2 hours at -20°C and the pellet containing the nucleus was stored in ice. Samples were then centrifuged at 18,000 x g at 4°C for 15min, and the pellet was processed for RNA isolation using TRIzol/chloroform method. The nuclear fraction was washed thrice with ice- cold HLB buffer and centrifuged at 300 x g at 4°C for 2min and the pellet was stored at -80°C for further use. HIV-1 RNA (*gag*) abundance was quantified from cytoplasm at different time points as a measure of HIV-1 transcription using forward primer 5'- TTGTACTGAGAGACAGGCT -3' and Reverse primer 5'- ACCTGAAGCTCTCTTCTGG-3'.

## Nuclear run-on assay

A nuclear run-on assay was performed as described previously [86]. For nuclei isolation, JTAg cells were infected by NL4–3 zsGreen virus with/without the Repsox treatment (10µM) and harvested using ice-cold hypotonic solution (150mM KCl (Sigma), 4mM MgOAc (Sigma), and 10mM Tris-HCl (Sigma), pH 7.4) and were pelleted by centrifugation 300 x g for 5min. The cells were then lysed in lysis buffer (150mM KCl, 4mM MgOAc, 10mM Tris- HCl, pH 7.4, and 0.5% NP-40). The crude nuclei were mixed with 10mM ATP, CTP, GTP, and BrUTP and incubated at 28 °C for 5min in the presence of RNase inhibitor. Further, the RNA was isolated with TRIzol reagent. The nascent transcripts were then Immunoprecipitated by anti-BrdU antibody (Merck, Cat# B8434) and converted to cDNA for qPCR analysis. qPCR analysis for gag was done using forward 5'- TTGTACTGAGAGACAGGCT -3' and Reverse 5'- ACCTGAAGCTCTCTTCTGG-3' primers.

## LTR luciferase activity

For the LTR-Luc experiment, LTRs from NL4–3 and various TF viruses were cloned into the PGL3-Luc basic vector (Promega) [17]. To analyze the effect of TGF-β across different HIV LTRs, LTR-Luc (100ng) constructs, along with Vpr-expressing pCDNA-HA-Vpr (50ng), and pCDNA-Tat (10ng) were transfected into HEK293T cells in a 12-well plate and challenged with DMSO or TGF-β inhibitors Repsox (10 µM) or SB525334 (10 µM). After 24 hours, the luciferase assay was performed to assess the effect of TGF-β receptor inhibition on LTR activity. SRSF1 was used as a positive control for the assay, based on our previous findings [17]. For assessing LTR-driven transcriptional activity in CD4+T cells were infected at an equal MOI of 5, using NL4–3 Env−R−Luc and NL4–3 Env−R+Luc [22,87] reporter viruses pseudotyped with VSVG. Luciferase activity was measured 24 hours post-infection. Data were normalized to total protein quantified by the Bradford assay.

## Immunoblotting

For immunoblotting, samples were lysed using RIPA lysis buffer or with DDM lysis buffer (100 mM NaCl, 10 mM HEPES [pH 7.5], 50 mM TCEP, 1% *n*-dodecyl-β-d-maltoside [DDM]) with 2X PIC (protease-inhibitor cocktail) and 2 mM sodium orthovanadate, 1 mM NaF and rocked on ice for 30 min. Following this, the lysates were clarified by centrifugation at 10,000 × *g* for 15 min, and the supernatant was collected and mixed with 4 × Laemmli buffer with 50 mM TCEP. Samples were either run on 8% or 15% or 12% tricine gels, depending upon the protein size. Next, gels were electroblotted on the PVDF membrane (Immobilon-FL, Merck-Millipore). The membrane was blocked using a commercial blocking buffer (BIORAD) or odyssey blocking buffer for 15 min, followed by primary and secondary antibody incubations. Post antibody incubation membrane was washed thrice (5 min per wash) using Tween20 containing 1x Tris-Buffered saline (TBST). Details of antibodies are provided in the S1 Table. Full blown immunoblots are provided in S1 File.

## Chromatin immunoprecipitation (ChIP)

For Chromatin Immunoprecipitation, 10 million JTAg cells or 10 million PHA/IL2 activated PBMCs (Product code#CL010) per condition were fixed with and crosslinked with 1% formaldehyde (Sigma) for 10 minutes. The reaction was quenched by adding 0.125 M glycine for 5 minutes at room temperature. The cells were rinsed three times in cold PBS before being suspended in a lysis buffer containing 50 mM HEPES-KOH (pH 7.5), 140 mM NaCl, 1 mM EDTA, 10% glycerol, 0.5% NP-40, 0.25% Triton X-100, with protease inhibitors. The nuclei were centrifuged at 800 × *g* for 5 minutes at 4 °C and then suspended in lysis buffer containing 10 mM Tris-HCl pH 8.0, 200 mM NaCl, 1 mM EDTA, 0.5 mM EGTA, with protease inhibitors and incubated on ice for 5 minutes. The nuclei were centrifuged at 800xg for 5 minutes at 4 °C and then suspended in a buffer consisting of 10 mM Tris-HCl pH 8.0, 200 mM NaCl, 1 mM EDTA, 0.5 mM EGTA, and protease inhibitors and incubated on ice for 10 minutes. The nuclei were gathered and reconstituted in a sonication buffer consisting of 10 mM Tri-HCl (pH 8.0), 100 mM NaCl, 1 mM EDTA, 0.5% EGTA, 0.1% Sodium deoxycholate, and 0.5% N-lauryl sarcosine, along with protease inhibitors. The DNA was fragmented using a probe sonicator for 15 cycles, with each cycle consisting of 30 seconds of sonication followed by 30 seconds of rest. The samples were centrifuged at 16000xg for 10 minutes at a temperature of 4 °C, after treatment with a 1% Triton X-100. For each ChIP cycle, a portion of sonicated DNA was subjected to reverse-crosslinking and then analyzed on a 1% agarose gel to verify the size of the fragments (300–500 bp). Immunoprecipitation was performed on chromatin samples (25 µg) by specific antibodies (anti-smad2/3, anti-Tat, anti-SRSF1, anti-RNAPol II), followed by overnight incubation at 4°C. Following an incubation period, 30 µl of Protein G beads (Invitrogen) were added and incubated for one hour at a temperature of 4 °C. The beads were sequentially washed for 3 minutes each in low salt (20 mM Tris-HCl pH 8.0, 150 mM NaCl, 2 mM EDTA, 0.1% SDS, 1% Triton X-100), high salt (20 mM Tris-HCl pH 8.0, 500 mM NaCl, 2 mM EDTA, 0.1% SDS, 1% Triton X-100), LiCl buffer (10 mM Tris-HCl pH 8.0, 0.25 M LiCl, 1% NP40, 1% Sodium deoxycholate), and TE buffer. The beads were washed with 150 µl of elution buffer containing 50 mM Tris-HCl pH 8.0, 10 mM EDTA, 1% SDS, and 50 mM NaHCO3. Then, reverse crosslinking was performed with 1 µl of RNaseA (1mg/ml) at a temperature of 37 °C for 30 minutes and subsequently digestion of proteins by adding 1 µl of proteinase K (20mg/ml) and allowing it to react at a temperature of 65°C for 4 hours. The DNA was eluted and used for qPCR analysis (Primers used for the quantification are provided in the S1 File).

## Immunofluorescence

HEK293T cells were seeded in 12 well at $1 \times 10^5$ cells/well with coverslips and allowed to adhere overnight. Cells were then transfected with vector or Vpr or indicated mutants for 20 h. Cells were then fixed in 4% PFA for 20 minutes at room temperature and permeabilized with 0.5% Triton X-100 in PBS for 15 min. Samples were then washed in 1 × PBS and incubated with blocking buffer (3% BSA, 0.05% Tween 20, and 0.04%NaN3 in PBS) for 30 min. Cells were probed with appropriate primary antibodies anti-HA Rabbit (CST), anti-γH2AX (Biolegend), and then washed in PBST (0.05% Tween

20 in PBS) thrice for 5mins each and probed with Alexa Fluor-conjugated secondary antibodies. Nuclei were stained with Hoechst 33342(Sigma).

### PBMC and CD4+ T cell culture

Human Peripheral Blood Mononuclear Cells (PBMCs, Product code CL010)tested negative for mycoplasma, yeast, fungi, bacteria, HBV, and HIV were purchased from HiMedia and grown and maintained in RPMI-1640 (Gibco) supplemented with 10% FBS or advanced RPMI supplemented with 2% FBS. The cells were stimulated with 5 µg/mL phytohemagglutinin (PHA, Sigma-Aldrich) and 50 IU/mL recombinant human IL-2 (Gibco). IL-2 was supplemented in each culture for the maintenance of the cells. CD3+/CD4+ T cells were purified by positive selection by magnetic separation with a CD4+ isolation kit (Miltenyi Biotec) from PBMCs as per the manufacturer's protocol. CD4+ T cells were cultured and counter-stained with anti-CD4-APC antibody (1:100) and anti-CD3-FITC labelled antibody (1:100) and analysed for its enrichment by flow cytometry. Antibodies were diluted in PBS/ 1% BSA/ 0.05% NaN3 (PBA). CD4+ T cells from three independent donors were used as reported in [17] supplemented with IL-2 for maintenance with Advanced RPMI 1640 with 2% FBS.

### Replication kinetics of WT HIV-1 NL4–3 (R+) and NL4–3 (R-) viruses in PBMCs

PHA/IL2 stimulated PBMCs were infected with NL4–3 (R+) [88] and NL4–3 (R-) (this study) produced by transfection of HEK293T cells and transiently pseudotyped with VSV-G. PBMCs were washed after 6hrs of infection and were equally divided and either challenged with Repsox(10µM) or DMSO. Infected PBMCs were collected every third day and processed for FACS analysis of intracellular p24. On day 6 cells as well as corresponding supernatants were also processed for ciTRAN level estimation and ELISA for cell-free TGFβ.

### ELISA for total TGF-β

For quantifying total TGF-β from the culture supernatant, 50 µl of the supernatant from all samples was used. The assay was performed as per the manufacturer's instructions and a standard curve was generated from the kit components of total TGF-β ELISA kit (catalogue#436707, LEGEND MAX Total TGF-β1 ELISA Kit).

### Cell viability assay

CD4+ T cells were treated with Repsox at 10µM. After 24 hours of compound addition, Alamar Blue reagent (HiMedia) was added to each well according to the manufacturer's protocol, and the plate was incubated at 37°C for 4 h. The absorbance was measured at 570nm using a SpectraMaxi3X plate reader.

### Conservation of amino acids across Vpr

We obtained alignment of Vpr amino acid sequences derived from patient DNA samples using the HIV LANL database (https://www.hiv.lanl.gov/). The alignments were analysed for conservation using Jalview software [89].

### Statistical analysis

Statistical analyses were performed using GraphPad Prism 9.0. Experiments were performed at least 3 times, and data were presented as the means±standard deviation (SD) unless stated otherwise in legends. The normality of the data was determined using Shapiro-Wilk test before choosing the test for statistical analysis. Multiple comparisons within a single or multiple groups were performed using one-way ANOVA or two-way ANOVA respectively. Following a one-way ANOVA, Dunnett's multiple comparison was performed, while A Dunnett's or Sidaks multiple comparison was performed following a Two-way ANOVA based on selective comparisons. Unpaired t-test was performed to compare two groups. All reported differences were ns=non-significant, *p<0.05, **p<0.01, ***p<0.001 and ****p<0.0001 unless otherwise stated.

## Supporting information

**S1 Fig. (A) HIV-1 Vpr (P05928) modelled using Expasy Swissprot representing the mutations studied.** Right panel-summarizing various functions of Vpr mutants: Green indicates proficient and Red indicating deficient in the depicted function. (B) Immunoblot showing packaging of HA-Vpr and its mutants in LVs produced from HEK293T. (C) RNase R treatment (30 mins) was given to RNA isolated from the Vpr or without transduced JTAg cells and levels of circ-*SMARCA5*, linear *SMARCA5*, *GAPDH* was assessed by qRT-PCR. All the data were normalized to control RNaseR samples (n = 3 ± SD). (D) Immunofluorescence of γH2X along with HA-Vpr WT and indicated mutants (Scale bar-100μm). (E) Immunoblot showing the DNA damage marker γH2X with different Vpr mutants with corresponding control as β-actin. (TIF)

**S2 Fig. (A) Knockdown efficiency of DCAF1 by shRNAs in JTAg cells quantified by qRT-PCR.** (B) Cell cycle analysis with G2/Mi inhibitor (Apegenin) and G1 inhibitor (CPI 203) and DMSO after 24 hours of treatment in JTAg cells. (C) Knockdown of ATM, ATR, DNAPK in JTAg cells was assessed by qRT-PCR. Data were normalized to *GAPDH* in each conditions (n = 3 ± SD). (TIF)

**S3 Fig. (A) qRT-PCR of ciTRAN (circ-*SMARCA5*) and Linear *SMARCA5* after transducing JTAg cell with Vpr (-), Vpr (+) LVs (n = 3 ± SD).** (B) ciTRAN levels were quantified (after 6 h) from CD4$^+$ T cells that received (20μM) of etoposide with or without Repsox (10μM) treatment for 4h. (C) Effect of Repsox on ciTRAN levels was assessed by qRT-PCR from HEK293T cells after 24 hours transfected with either vector or Vpr plasmid or Repsox (10μM) treatment. (D) Levels of QKI post-shRNA-mediated knockdown by qRT-PCR from JTAg cells. Data were normalized to GAPDH. The two-tailed Student's t-test (unpaired) or one-way ANOVA with Dunnett's Multiple comparison test was used to assess the significance between two or more groups, ns = non-significant, $p < 0.05$, $p < 0.01$, $p < 0.001$ and $p < 0.0001$. (TIF)

**S4 Fig. (A) Flow cytometry of CD3$^+$/ CD4$^+$ cells purified from PBMCs.** (B) TGF-β family protein comparison from uniport database as, P01137- TGF-β1, P61812- TGF-β2, P10600- TGF-β3 and aligned using clustal omega to determine their percent identity. (C) Analysis of TGFB superfamily members from human primary CD4$^+$ cells by qRT-PCR after transducing with LVs with or without Vpr. (D) Cell viability assay using Alamar blue with/without Repsox in human primary CD4$^+$ cells. (E) Conservation of indicated amino acid residues in Vpr sequences analyzed from the LANL database. (TIF)

**S5 Fig. (A) Representative flow-cytometry plots depicting the levels of intracellular p24 three days post-infection of PHA/IL2 stimulated PBMCs obtained from three different donors.** The cells were challenged with HIV-1 NL4–3 (R+) and NL4–3 (R-) viruses and cultured with or without Repsox (10 μM) or TGFβ (10ng/ml). (B) Luciferase activity assay performed after 24 hours following HIV-1 Luc Vpr(+/−) infection of CD4 + primary T cells. presented as biological replicates normalized to total protein quantified by Bradford assay. Two-way ANOVA with Dunnett's Multiple comparison test was used to assess the significance between two or more groups, ns = non-significant, $p < 0.05$, $p < 0.01$, $p < 0.001$ and $p < 0.0001$. (TIF)

**S1 Table. Plasmids and reagents used in this study.** (PDF)

**S1 File. Oligo sequences and Full western Blots.** (PDF)

**S1 Data. All raw values for all graphs represented in figures.** (XLSX)

## Acknowledgments

The authors are indebted to M. Pizzato, the NIH AIDS reagent program, and NIBSC for reagents and cell lines.

## Author contributions

**Conceptualization:** Ajit Chande.

**Data curation:** Aditi Choudhary, Katyayani Mallick, Rishikesh Dalavi, Ajit Chande.

**Formal analysis:** Aditi Choudhary, Katyayani Mallick, Rishikesh Dalavi, Ajit Chande.

**Funding acquisition:** Ajit Chande.

**Investigation:** Aditi Choudhary, Katyayani Mallick, Rishikesh Dalavi, Ajit Chande.

**Methodology:** Aditi Choudhary, Katyayani Mallick, Rishikesh Dalavi.

**Project administration:** Ajit Chande.

**Resources:** Ajit Chande.

**Software:** Aditi Choudhary, Rishikesh Dalavi.

**Supervision:** Ajit Chande.

**Validation:** Aditi Choudhary, Katyayani Mallick, Rishikesh Dalavi.

**Writing – original draft:** Aditi Choudhary, Ajit Chande.

**Writing – review & editing:** Ajit Chande.

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
