## [Decision Letter · Decision Letter 0]

Dear Dr Chande,

Thank you very much for submitting your manuscript "HIV-1 Vpr triggers TGF-β signaling which promotes the expression of proviral circular RNA ciTRAN during infection" for consideration at PLOS Pathogens. As with all papers reviewed by the journal, your manuscript was reviewed by members of the editorial board and by several independent reviewers. In light of the reviews (below this email), we would like to invite the resubmission of a significantly-revised version that takes into account the reviewers' comments.

The referees acknowledge that the findings show that Vpr induces the expression of circular RNA ciTRAN via a TGFbeta-dependent mechanism. However, they raise a number of points that represent a substantial amount of work.

Specifically, the use of the Q65R mutant seems inappropriate to point out a DNA damage defect, as this Vpr mutant is deficient for several Vpr’s activities. The analysis of Vpr mutants in Figure 1 does not indicate that the DDR pathway is central to the induction of ciTRAN expression by Vpr (even if cTRAN is induced by DNA damage agents). Referring to Vpr-wt as the DNA-proficient Vpr and to Vpr Q65R to the DNA damage-deficient Vpr, is misleading.

More importantly, as stated by two reviewers, it is unclear whether the modulation of TGF-beta signalling is linked to the activity of Vpr.

Thus, I would like to invite you to submit a revised manuscript with the understanding that the referee concerns must be fully addressed and their suggestions taken on board. In addition, it is mandatory to compare wt Vpr and delta-Vpr viruses in the different settings to show that a delta-Vpr virus is not sensitive to Repsox or other TGF-beta inhibitors in Figure 5 and to show that ciTRAN increase is due to Vpr-driven increase in TGFbeta expression in Figure 6.

In addition, we feel the paper is difficult to follow for a broad readership. The authors need to put their data in the context of what is known about HIV-1 Vpr.

We cannot make any decision about publication until we have seen the revised manuscript and your response to the reviewers' comments. Your revised manuscript will be sent to reviewers for further evaluation.

Sincerely,

Florence Margottin-Goguet

Academic Editor

PLOS Pathogens

Susan Ross

Section Editor

PLOS Pathogens

Michael Malim

Editor-in-Chief

PLOS Pathogens

orcid.org/0000-0002-7699-2064

The referees acknowledge that the findings show that Vpr induces the expression of circular RNA ciTRAN via a TGFbeta-dependent mechanism. However, they raise a number of points that represent a substantial amount of work.

Specifically, the use of the Q65R mutant seems inappropriate to point out a DNA damage defect, as this Vpr mutant is deficient for several Vpr’s activities. The analysis of Vpr mutants in Figure 1 does not indicate that the DDR pathway is central to the induction of ciTRAN expression by Vpr (even if cTRAN is induced by DNA damage agents). Referring to Vpr-wt as the DNA-proficient Vpr and to Vpr Q65R to the DNA damage-deficient Vpr, is misleading.

More importantly, as stated by two reviewers, it is unclear whether the modulation of TGF-beta signalling is linked to the activity of Vpr.

Thus, I would like to invite you to submit a revised manuscript with the understanding that the referee concerns must be fully addressed and their suggestions taken on board. In addition, it is mandatory to compare wt Vpr and delta-Vpr viruses in the different settings to show that a delta-Vpr virus is not sensitive to Repsox or other TGF-beta inhibitors in Figure 5 and to show that ciTRAN increase is due to Vpr-driven increase in TGFbeta expression in Figure 6.

In addition, we feel the paper is difficult to follow for a broad readership. The authors need to put their data in the context of what is known about HIV-1 Vpr.

Reviewer's Responses to Questions

**Part I - Summary**

Reviewer #1: In their manuscript, Choudhary et al. describe the activation of the TGF-beta signal pathway by HIV Vpr, which in turn activates the expression of the proviral circular RNA ciTRAN.

Within their manuscript, the authors build up on their previous work and show that HIV Vpr, upon infection and transfection, upregulates TGF-beta signaling, which is controlled by the use of a TGFBR1 inhibitor or genetic ablation. TGF upregulation in turn promotes SMARCA5 promoter activity, which is shown by luciferase reporter assays. Enhanced SMARCA5 activity leads to higher ciTRAN RNA expression levels, which acts proviral. Blocking the Vpr-induced TGF-beta signaling pathway leads to a reduction in HIV infectivity (Fig 6). In general, the manuscript is well written and data in the figures are clearly and logically presented. Most experiments include the necessary controls and, in addition to cell lines and HIV lab strains, primary cells and transmitted founder viruses were used.

Reviewer #2: In this manuscript, Choudhary et al present a series of experiments meant to demonstrate that HIV-vpr-driven induction of ciTRAN is mediated by vpr-driven upregulation of TGF�� and TGF-� signaling and that TGF-� impact on HIV infection is mediated by ciTRAN. The manuscript is relatively well-written, but there are a lot of missing or contradicting information especially in the legends and methods. Moreover, while the data implicating the vpr-activated DDR pathway in the increase of ciTRAN and demonstrating that TGF-� increases ciTRAN in HEK cells appear convincing, the data connecting vpr expression to TGF-� induction and the fact that ciTRAN increase is due to this vpr-driven increase in TGF-� expression are less convincing particularly in primary cells.

Moreover, the role of the TGF-� pathway in upregulating ciTRAN in primary cells is not convincing nor are the data meant to demonstrate that ciTRAN is mediating the effect of TGF-� on infection in primary cells. Considering that, in Jurkat cells, the AKT and PtdIns(3,4,5)P3 are, respectively, constitutively phosphorylated and increased compared to primary cells (PMID 1095869) and that these are among the target of TGF-� signaling in primary T cells (Cattley JBC 2020), recapitulating the effects of TGF-� in primary T cells is essential to the relevance of the reported findings to physiological systems.

This is especially true since the effect of TGF-� on infection in both primary T cells and THP-1 cells can be explained by the well-known TGF-driven upregulation of CCR5 in these cells.

Reviewer #3: In the MS the authors follow on from a previous paper in Sci Adv last year. They present data that the HIV-1 accessory protein Vpr induces the expression of a circular RNA ciTRAN via a TGFb-dependent mechanism initiated by Vpr within the incoming viral particle. This appears to be independent of DNA-damage and cell cycle arrest by Vpr, but is abolished by the Q65R mutation. The authors further show that inhibition of TGFb signalling reduces viral transcription.

The paper presents clear data that in experimental systems Vpr induces ciTRAN via a TGFb-dependent mechanism. However, the mechanistic and virological phenotypes are less compelling at the moment. The paper is also quite difficult to read; experiments and reagents are not well-described, and even by having the authors’ previous paper open at the same time, I struggled. Furthermore, there is not much attempt to put these data in the context of the known biology of Vpr, meaning the relevance of these observations to the pathophysiology of HIV/AIDS is unclear. In addition to the specific comments below, I recommend a wholesale revision of the methods and figure legends to clearly specify what construct/virus is being used where, and how much of it.

**Part II – Major Issues: Key Experiments Required for Acceptance**

Reviewer #1: (No Response)

Reviewer #2: - In most figures only 3 replicates are shown. According to the statistical paragraph these 3 replicates are technical and not biological replicates. Biological replicates should be shown and used for statistical analysis. T-test is not appropriate for this small sample size. A non-parametric test is needed for these comparisons unless the authors can demonstrate normality of their dataset (which is very hard with an n of 3).

- The experiments in Fig 3A are critical to demonstrate that TGF-�-signaling is initiated following Vpr expression. In these experiments it appears that transfection with WT Vpr stimulates TGF-� production by HEK293T cells which then increase luciferase expression following SMAD activation. Important controls such as demonstration of increased TGF-�1 levels in these cultures and/or blocking of TGFBR1 or of TGF-� in the WT-vpr condition are missing. Moreover, in Fig 3D pSMAD increase seems to be present in both infected (vpr+) and uninfected cells, suggesting that TGF-� increase may be a bystander effect of inflammation in infected cells that affects the entire culture. Also levels of total SMAD are not shown or used to normalize nor it is stated if the flow data are an example of several replicates.

- Along the same lines the experiments primary cells, according to the figure legends, were performed with a viral construct including several proteins other than just Vpr and not Vpr only as shown in the figure. Hence, we can conclude that viral infection, but not necessarily Vpr, drives higher levels of TGF-�. Lentiviral particles appropriately characterized (see minor issues below) and expressing only Vpr should be used in primary cells to demonstrate Vpr-specific increase in TGF-� in primary cells and data from several biological replicates should be used for the analysis. Possibly ChiPseq of SMAD2/3 should be also used in primary cells following VPR-LV to demonstrate that VPR engages the TGF-� pathway in primary cells and that is responsible for increased activation of the SMARAC5 promoter.

Reviewer #3: Specific points:

• The authors refer to the Vpr Q65R mutant as a “DNA damage-deficient mutant”. While technically true, that misrepresents what this mutant does. Q56R is deficient for interaction with DCAF1 and so cannot recruit the Cullin-4 E3 Ligase complex. This mutant is deficient for all Vpr functions as far as we know. It also implies that something is being targeted in Ub-dependent manner to induce the TGF-beta-based induction of ciTRAN. This is fundamental to the biology and hasn’t been addressed or discussed. It would be important to look at whether DCAF1/Cul4 knockdown abolishes Vpr’s activity. Also, a number of high profile targets of Vpr are known. It would improve the general interest of the paper if some of these could be ruled in and out.

• There seems to be a disconnect between the authors assertion that Vpr-mediated induction of a TGFb-dependent upregulation of ciTRAN is DNA damage independent and the effects of DNA damaging agents or DSBs produced by Cas9. This interpretations is based on the lack of reduction of the Vpr phenotype by inhibitors of ATM, ATR or DNA-PK. The authors didn’t include Etop/Doxirubicin/Cas9 in the latter experiments making this conclusion uninterpretable. More properly the effect is independent of the canonical response to DNA damage, not necessarily the damage itself.

• The virological experiments in figs 5 and 6 are difficult to interpret. As I understand the virus they are using, it is an env-defective NL4.3 with ZcsGreen in the Nef ORF. If so this expresses Vpr endogenously, but they are the packaging Vpr in trans on top of this. I guess this must be a mistake and the virus is Vpr-defective but this highlights the lack of clarity for this reviewer.

• I’m unclear why the authors have not compared full replication of the Vpr+ and Vpr- viruses in primary activated T cells. While the authors suggest that viral gene expression is reduced in the absence of Vpr (or treatment with TGFRi), its well known that the difference in replication between Vpr+ and – mutants in cultured primary T cells is marginal at best, and usually non-existent. How does the treatment with Respox compare to the multiround replication of Vpr mutant viruses?

• The data from HIV-1 infected individuals in Fig 6 is based on whole PBMC transcriptomics. This cannot be compared to the in vitro data. The majority of the cells at a given time in the circulation are not infected, so unless the authors can prove they were exposed to Vpr, it is likely that the general upregulation of TGFb is a different effect.

• Lastly the authors need to put their data intot he context of what is known about Vpr-defective mutants of HIV-1. For example, how much of the Vpr-induced global transcriptional change is TFGb-dependent (Bauby et al 2021); is Vpr suppression of inflammatory responses downstream of cGAS required for this activity; can TGFb induction explain the permissivity of resting cells and macrophages to Vpr+ viruses; and looking back to the authors previous paper, in infected primary T cells can they show that ciTRAN is responsible.

**Part III – Minor Issues: Editorial and Data Presentation Modifications**

Reviewer #1: 1. Better explain DNA Damage and Q65 mutant.

Although most of the results in the manuscript complement each other and nicely explain how Vpr, TGF-beta, and ciTRAN are connected, the DNA damage part of the story seems more complicated and is not formulated straight forward.

The authors show that Etoposide is inducing DNA damage and activates ciTRAN, similar to WT but not Q65R Vpr, but the fact that blocking DDR factors are not involved in ciTRAN regulation suggests that the effect of Vpr is DNA-damage independent.?! This is very well possible but needs to be explained more clearly. Is TGF-beta signaling induced in the presence of Etoposide?

Vpr Q65R fails to interact with its E3 Ligase complex/VprBP, so it is conceivable that Q65R is not inducing the degradation or modification of an unknown factor responsible for signaling/ciTRAN upregulation?

2. Literature

It has recently been shown that the VprBP protein is mitigating TGF-beta signaling, maybe this phenotype is connected and the manuscript should be discussed?

DOI: 10.1093/jmcb/mjz057

Also, recently Vpr has been shown to target TGIF2 via miRNA, maybe this paper should be discussed as well in light of the findings of the authors? Is this phenotype connected?

DOI: 10.1371/journal.pone.0261971

3. HIV pseudotypes /LV particles with Vpr Q65R.

What happens to TGF-beta induction and ciTRAN levels upon infection with HIV harboring Vpr-Q65R?

Reviewer #2: The data from mutant vpr in Fig 1B are also included in FigS9G of the previous Science Advances publication. However, the triplicates included in the current manuscript are very different from those included in the previous publication.

The impact of TGF-� on virus replication in T cells was incorrectly quoted in lines 75-76 as enhancing viral replication while the papers cited either demonstrate enhanced infection by CCR5-tropic viruses due to TGF-�-driven increased expression of CCR5 (ref 38 – also reported in the context of myeloid cells) or inhibition of viral replication (opposite effect) and latency reversal (ref 39)

There is no information on the quantity of vpr in lentiviral particles used for the various experiments and in particular for the data in Fig 1B. How were the different mutant particles assessed and normalized to each other for VPR content? Is it possible that mutation change the efficiency of incorporation and since the amount of ciTRAN is likely proportional to the amount of vpr, this control is important especially in light of contrasting data from previous publication in Science Advances (Fig S9G and Fig1 in the current manuscript). A supplemental figure with data demonstrating similar levels of vpr incorporation should be included.

In Figure 5B are these TF full length molecular clones or just the LTR followed by luc? It seems the latter since the data are expressed as % of luc activity, but the text mentions TF molecular clones and there is no info in the methods. This is important because the LTR only vectors don’t have vpr which makes the interpretation of the data confusing.

The experiments in primary cells have contradicting information. In the methods TGF-� ELISA appears to have been done with PBMC, but in the legend CD4+ T cells are used. CD4 T cells appear to have been isolated via positive selection. I am not aware of a clone that is not blocked by the isolation kits Abs, but there is no clone info. Also, what does it mean that CD4 were isolated from PBMC pooled from 3 donors? That does not mean that the experiment was done with cells from 3 different donors as it should be.

There is no information on the TGF-� - ELISA used (manufacturer name etc)

Reviewer #3: • In Fig 4a Tat IP is not pulling down the viral DNA specifically. Tat binds RNA so it must be doing it indirectly through the nascent transcript.

PLOS authors have the option to publish the peer review history of their article (what does this mean? ). If published, this will include your full peer review and any attached files.

**Do you want your identity to be public for this peer review?** For information about this choice, including consent withdrawal, please see our Privacy Policy .

Reviewer #1: No

Reviewer #2: No

Reviewer #3: **Yes: ** stuart neil
---

## [Decision Letter · Decision Letter 1]

PPATHOGENS-D-24-01887R1

TGF-β signaling induced by HIV-1 Vpr promotes the expression of proviral circular RNA ciTRAN

PLOS Pathogens

Dear Dr. Chande,

Thank you for submitting your manuscript to PLOS Pathogens. After careful consideration, we feel that it has merit but does not fully meet PLOS Pathogens's publication criteria as it currently stands. Therefore, we invite you to submit a revised version of the manuscript that addresses the points raised during the review process.

Please submit your revised manuscript within 60 days (April 17). If you will need more time than this to complete your revisions, please reply to this message or contact the journal office at plospathogens@plos.org. Please include the following items when submitting your revised manuscript:

We look forward to receiving your revised manuscript.

Kind regards,

Florence Margottin-Goguet

Academic Editor

PLOS Pathogens

Susan Ross

Section Editor

PLOS Pathogens

 Sumita Bhaduri-McIntosh

Editor-in-Chief

PLOS Pathogens

orcid.org/0000-0003-2946-9497

 Michael Malim

Editor-in-Chief

PLOS Pathogens

orcid.org/0000-0002-7699-2064

**Additional Editor Comments:**

We would like the authors to play down the role of TGFbeta on HIV replication and focus on the data on Vpr-induced TGFbeta release and ciTRAN upregulation. Please, return a new version by taking into account Reviewer 2 comments.

**Reviewers' Comments:**

Reviewer's Responses to Questions

**Part I - Summary**

Reviewer #1: (No Response)

Reviewer #2: This revised manuscript is much improved in terms of clarity. However, the data in primary CD4 T cells (figure 6) still do not support the relevance of the ciTRAN upregulation by VPR-mediated TGFb plays a major role in shaping HIV replication in primary CD4 T cells. The authors present strong data in support of vpr-driven increase in TGFb, which in turn upregulates transcription of ciTRAN. However, the connection with primary HIV infection and the role of exogenous TGFb is weak at best. The levels of HIV replication is similar in RepSox-treated cells infected with Vpr+ or Vpr- virus (Fig 6H). At the same time, the Vpr+ virus seems to replicate slightly better despite the notable increase in TGFb and ciTRAN. Moreover, these experiments were performed only once in 1 donor and presented as technical replicates.

Finally, all the experiments in which HIV “infection” (or LTR activity) seems to be decreased by the TGFb inhibition , do not show the effect of exogenous TGF-b. Rather, they show a decrease that may be linked to vpr-induced increase in TGFb. If this is the case, this should be made clearer.

In some cases the authors seem to suggest that vpr is increasing TGFb production (which seems to be supported by the data), but the main message appears to be that vpr induces TGFb “signaling” which leads to some confusion.

Reviewer #3: the authors have addressed my major concerns adequately

**Part II – Major Issues: Key Experiments Required for Acceptance**

Reviewer #1: (No Response)

Reviewer #2: Experiments with primary CD4 T cells would have to demonstrate that in presence of vpr there is a significant downregulation of HIV replication when TGFb is inhibited. Additionally, they would have to demonstrate that exogenous TGFb increases HIV replication in primary CD4 T cells (both infection and transcription) and this is at least significant in vpr-negative infections (where supposedly there is no vpr-mediated TGFb upregulation). Finally, the activity of the LTR should be probed in presence of exogenous TGFb. Alternatively, the authors could modify the manuscript to focus on the fact that vpr-induced TGFb production leads to ciTRAN upregulation and that this mechanism may play a role in modulating HIV replication. Basically, they need to make it clear that the relevance of this to primary HIV replication is context dependent and may not be playing a role at all. They should also eliminate the analysis of clinical data correlating the residual plasma viral load with TGFb levels, because that can be indirect. Residual viral replication is more likely leading to residual immune activation and increased levels of TGFb rather than the other way around as suggested by the authors. The data in the context of the other data presented can be misleading.

Reviewer #3: (No Response)

**Part III – Minor Issues: Editorial and Data Presentation Modifications**

Reviewer #1: The authors adressed the points raised by this reviewer.

Reviewer #2: The test for normality of data is not listed in the statistical paragraph and there is no mention of correction for multiple comparisons or post-hoc tests in the cases in which ANOVA was used. The statistical paragraph mentions that the experiments were repeated at least 3 times, but the data shown in Fig 6 with primary T cells are from “technical triplicates”

Reviewer #3: (No Response)

PLOS authors have the option to publish the peer review history of their article (what does this mean? ). If published, this will include your full peer review and any attached files.

**Do you want your identity to be public for this peer review?** For information about this choice, including consent withdrawal, please see our Privacy Policy .

Reviewer #1: No

Reviewer #2: No

Reviewer #3: No

**Figure resubmission:**
---

## [Decision Letter · Decision Letter 2]

PPATHOGENS-D-24-01887R2

HIV-1 Vpr orchestrates ciTRAN upregulation through TGF-β induction

PLOS Pathogens

Dear Dr. Chande,

Thank you for submitting your manuscript to PLOS Pathogens. After careful consideration, we feel that it has merit but does not fully meet PLOS Pathogens's publication criteria as it currently stands. Therefore, we invite you to submit a revised version of the manuscript that addresses the points raised during the review process.

Please submit your revised manuscript within 30 days Aug 09 2025 11:59PM. If you will need more time than this to complete your revisions, please reply to this message or contact the journal office at plospathogens@plos.org. Please include the following items when submitting your revised manuscript:

We look forward to receiving your revised manuscript.

Kind regards,

Susan R. Ross, PhD

Section Editor

PLOS Pathogens

Susan Ross

Section Editor

PLOS Pathogens

Sumita Bhaduri-McIntosh

Editor-in-Chief

PLOS Pathogens

orcid.org/0000-0003-2946-9497

Michael Malim

Editor-in-Chief

PLOS Pathogens

orcid.org/0000-0002-7699-2064

**Additional Editor Comments:**

We would like you to revise the manuscript by responding to the criticisms of reviewer 2.

**Reviewers' Comments:**

Reviewer's Responses to Questions

**Part I - Summary**

Reviewer #2: The manuscript is partially responsive to the reviewers critiques, but there are important and confusing issues due to experiments listed and discussed for the reviewers but not included in the final version of the manuscript. Specifically:

In their response the authors mention a Fig6K and Fig6H, but these panels are not present in Figure 6 and not mentioned in the results. These results include that “while Vpr-driven TGF-β production is reversed by Repsox, infection-induced TGF-β levels remain unaffected (see revised Fig. 6K as fig1)” but the authors didn’t include this in the manuscript which actually has a Fig 6G which shows that there is no TGF-β production in the vpr- challenged cells. Hence, it is still unclear if vpr increases TGF-β levels (the only truly supporting data in this regard is the small transcriptional increase in vpr+ but not vpr- cells in Fig S4 )

Also, the findings that the addition of TGF-β fails to rescue the replication defect of the vpr deficient virus is critical to the overall interpretation of the results, but it was not added in the main manuscript. This is also shown in Fig3 for the reviewers, where exogenous TGF-β fails to rescue the replication of the R- virus and actually also shows that, although RepSox inhibits replication from the LTR in vpr+ virus, so does exogenous TGF-β to a very similar degree.

**Part II – Major Issues: Key Experiments Required for Acceptance**

Reviewer #2: (No Response)

**Part III – Minor Issues: Editorial and Data Presentation Modifications**

Reviewer #2: The authors need to incorporate the new findings and data mentioned to the reviewers in the manuscript and resolve the apparent contradiction between those data and the way the results are currently presented and described in figure 6.

PLOS authors have the option to publish the peer review history of their article (what does this mean? ). If published, this will include your full peer review and any attached files.

**Do you want your identity to be public for this peer review?** For information about this choice, including consent withdrawal, please see our Privacy Policy .

Reviewer #2: No

**Figure resubmission:**
---

## [Editor Report · Decision Letter 3]

Dear Dr Chande,

We are pleased to inform you that your manuscript 'HIV-1 Vpr orchestrates ciTRAN upregulation through TGF-β induction' has been provisionally accepted for publication in PLOS Pathogens.

Best regards,

Florence Margottin-Goguet

Academic Editor

PLOS Pathogens

Susan Ross

Section Editor

PLOS Pathogens

Sumita Bhaduri-McIntosh

Editor-in-Chief

PLOS Pathogens

orcid.org/0000-0003-2946-9497

Michael Malim

Editor-in-Chief

PLOS Pathogens

orcid.org/0000-0002-7699-2064
---

## [Editor Report · Acceptance letter]

Dear Dr Chande,

We are delighted to inform you that your manuscript, "HIV-1 Vpr orchestrates ciTRAN upregulation through TGF-β induction," has been formally accepted for publication in PLOS Pathogens.

Best regards,

Sumita Bhaduri-McIntosh

Editor-in-Chief

PLOS Pathogens

orcid.org/0000-0003-2946-9497

Michael Malim

Editor-in-Chief

PLOS Pathogens

orcid.org/0000-0002-7699-2064